# OptEx: Expediting First-Order Optimization with Approximately Parallelized Iterations

**Yao Shu**[#†]**, Jiongfeng Fang**[‡]**, Ying Tiffany He**[‡]**, Fei Richard Yu**[‡§]

[†]Guangdong Lab of AI and Digital Economy (SZ), China
[‡]College of Computer Science and Software Engineering, Shenzhen University, China
[§]School of Information Technology, Carleton University, Canada

## Abstract

First-order optimization (FOO) algorithms are pivotal in numerous computational domains, such as reinforcement learning and deep learning. However, their application to complex tasks often entails significant optimization inefficiency due to their need of many sequential iterations for convergence. In response, we introduce *first-order optimization expedited with approximately parallelized iterations* (OptEx), the first general framework that enhances the optimization efficiency of FOO by leveraging parallel computing to directly mitigate its requirement of many sequential iterations for convergence. To achieve this, OptEx utilizes a kernelized gradient estimation that is based on the history of evaluated gradients to predict the gradients required by the next few sequential iterations in FOO, which helps to break the inherent iterative dependency and hence enables the approximate parallelization of iterations in FOO. We further establish theoretical guarantees for the estimation error of our kernelized gradient estimation and the iteration complexity of SGD-based OptEx, confirming that the estimation error diminishes to zero as the history of gradients accumulates and that our SGD-based OptEx enjoys an effective acceleration rate of $\Theta(\sqrt{N})$ over standard SGD given parallelism of $N$, in terms of the sequential iterations required for convergence. Finally, we provide extensive empirical studies, including synthetic functions, reinforcement learning tasks, and neural network training on various datasets, to underscore the substantial efficiency improvements achieved by OptEx in practice. Our implementation is available at https://github.com/youyve/OptEx.

## 1 Introduction

First-order optimization (FOO) algorithms, such as stochastic gradient descent (SGD) [1], Nesterov Accelerated Gradient (NGA) [2], AdaGrad [3], Adam [4] etc., have already been the cornerstone of many computational disciplines, driving advancements in areas ranging from reinforcement learning [5] to machine learning [6]. These algorithms, which are widely known for their straightforward form of iterative gradient-based updates, are fundamental in solving both simple and intricate optimization problems. However, their applications usually encounter substantial optimization inefficiency, especially when addressing complex functions that not only are *expensive in evaluating their function values and gradients* but also necessitate *a large number of sequential iterations to converge* in practice, e.g., deep reinforcement learning [7] and neural network training [8].

To this end, parallel computing has been widely used in the literature to considerably enhance the *optimization (e.g., time) efficiency* of FOO by reducing the evaluation cost of function and gradient *per iteration* in FOO [9]. For instance, in the field of neural network training, techniques that are based

---

# Correspondence to: Yao Shu <shuyao@gml.ac.cn>

38th Conference on Neural Information Processing Systems (NeurIPS 2024).

on parallel computing, e.g., data parallelism [8, 10, 11, 12], model parallelism [13], and pipeline parallelism [14, 15], have been employed to reduce the evaluation time of loss function and parameter gradient by processing multiple input samples and network components concurrently. However, to the best of our knowledge, few efforts have been devoted to leveraging parallel computing to reduce the *number of sequential iterations* required for convergence to mitigate the optimization inefficiency in FOO. Different from the methods of reducing the evaluation time per iteration during optimization, which requires specialized human efforts in a specific domain (e.g., neural network training), the reduction of sequential iterations is likely to be more general since no such specialized domain efforts are required and thus shall enjoy a wider application in practice. This underscores the need to explore the potential of parallelizing sequential iterations in standard FOO.

However, the inherent iterative dependency in FOO where the output of each iteration servers as the input of the next iteration, poses a significant barrier to independent and concurrent iteration execution, thereby making it nearly impossible to realize iteration parallelism within standard FOO. To this end, we develop a novel framework called *first-order optimization expedited with approximately parallelized iterations* (OptEx) that is capable of bypassing the challenge of inherent iterative dependency in standard FOO and therefore make parallelized iterations in FOO possible. Specifically, our framework begins with a novel kernelized gradient estimation strategy, which uses the history of gradients during optimization to predict the gradients for any input within the domain such that these estimated gradients can be used in standard FOO algorithms to determine the inputs for the next few iterations. We further introduce the techniques of separable kernel function and local history of gradients to enhance the computational efficiency of this gradient estimation (Sec. 4.1). We then apply standard FOO algorithms with this kernelized gradient estimation to determine the inputs for the next $N$ sequential iterations efficiently (namely proxy updates), aiming to approximate the ground-truth sequential updates and bypass the iteration dependency in standard FOO (Sec. 4.2). Lastly, we complete our approximately parallelized iterations for standard FOO by leveraging parallel computing with parallelism of $N$ to concurrently execute standard FOO algorithms over these $N$ inputs obtained from our proxy updates using the ground-truth gradients (Sec. 4.3).

Apart from proposing our innovative OptEx framework, we further establish rigorous theoretical guarantees and extensive empirical studies underpinning its efficacy. Specifically, we give a theoretical bound for the estimation error of our kernelized gradient estimation. Remarkably, this error approaches zero asymptotically as the number of historical gradients increases, ranging across a broad spectrum of kernel functions. This suggests that our kernelized gradient estimation can facilitate effective proxy updates to help parallelize sequential iterations in FOO (Sec.5.1). Building on this, we delineate both upper and lower bounds for the sequential iteration complexity of our SGD-based OptEx, showing that our SGD-based OptEx is able to reduce the sequential iteration complexity of standard FOO algorithms at a rate of $\Theta(\sqrt{N})$ with parallelism of $N$ (Sec.5.2). Finally, through extensive empirical studies, including the optimization of synthetic functions, reinforcement learning tasks, and neural network training on both image and text datasets, we demonstrate the consistent advantages of our OptEx in expediting existing FOO algorithms (Sec. 6).

To summarize, our contribution to this work includes:

- To the best of our knowledge, we are *the first to develop a general framework* (i.e., OptEx) that can leverage parallel computing to approximately parallelize the sequential iterations in FOO, thereby considerably reducing the sequential iteration complexity of FOO algorithms.
- We provide *the first upper and lower iteration complexity bound* for SGD-based OptEx, which gives an effective acceleration rate of $\Theta(\sqrt{N})$ with parallelism of $N$.
- We conduct extensive empirical studies, including the optimization of synthetic function, reinforcement learning tasks, and neural network training on both image and text datasets, to support the efficacy of our OptEx framework.

## 2   Related Work

**Reduction of Iteration Complexity.**   In the literature, various techniques have been developed to enhance the optimization efficiency of FOO by improving their sequential iteration complexity. For example, variance reduction strategies [16, 17, 18] have been proposed to accelerate stochastic optimization by effectively reducing the gradient variance and therefore aligning the iteration complexity of SGD with that of gradient descent (GD) in expectation. These strategies usually yield significant

improvements in high-variance problems whereas their compelling performance is hard to extend to low-variance scenarios and deterministic contexts. Meanwhile, adaptive gradient methods, e.g., AdaGrad [3], Adam [4], and AdaBelief [19], have been introduced to employ an adaptive learning rate for a better-performing optimization where fewer iterations are required for convergence. Furthermore, acceleration techniques like the Nesterov method [2] and momentum-based updates [20] have also been proven to be capable of reducing the sequential iteration complexity for GD and SGD efficiently. *Orthogonal to these established methodologies, our paper introduces parallel computing as a distinct and innovative strategy to further decrease the sequential iteration complexity of FOO. Of note, such an approach not only stands independently but also offers potential for synergistic integration with existing methods, promising enhanced optimization outcomes.*

**Reduction of Time Complexity Per Iteration using Parallel Computing.** In the realm of enhancing the computational efficiency of FOO, parallel computing has emerged as a rescue by reducing the time complexity per iteration in FOO. Particularly in the field of neural network training, data parallelism [8, 10, 11, 12] has been introduced to evaluate the gradients of model parameters w.r.t mini-batch input samples simultaneously. In addition to data parallelism, model parallelism [13] has been developed to process various neural network components concurrently. Furthermore, pipeline parallelism [14, 15] divides the neural network into stages and assigns each stage to a different device, allowing different stages of the computation to be executed in parallel across the pipeline. However, the tailored nature of these methods constrains their application to wider contexts. *Contradictory to these case-specified solutions, this paper proposes a general framework that can leverage parallel computing to enhance the optimization efficiency of FOO in wide practical applications.*

## 3 Problem Setup

In this paper, we aim to enhance the optimization efficiency of the following stochastic minimization problem by leveraging parallel computing with parallelism of $N$:

$$\min_{\boldsymbol{\theta} \in \mathbb{R}^d} F(\boldsymbol{\theta}) \triangleq \mathbb{E}\left[f(\boldsymbol{\theta})\right] . \tag{1}$$

Here, $\nabla f(\boldsymbol{\theta})$ is assumed to follow a specific Gaussian distribution, i.e., $\nabla f(\boldsymbol{\theta}) \sim \mathcal{N}(\nabla F(\boldsymbol{\theta}), \sigma^2 \mathbf{I})$ for any $\boldsymbol{\theta} \in \mathbb{R}$, which has already been widely used in the literature [21, 22, 23]. Besides, we adopt a common assumption that $\nabla F$ is sampled from a Gaussian process, i.e., $\nabla F \sim \mathcal{GP}(\mathbf{0}, \mathbf{K}(\cdot, \cdot))$ [24, 25, 26]. Of note, (1) has found extensive applications in practice, e.g., neural network training [27] and reinforcement learning [28]. Importantly, although our primary focus is on this stochastic optimization, our method can also be applied to deterministic optimization (evidenced in Sec. 6.1).

Standard FOO algorithms commonly optimize (1) in an iterative and sequential manner:

$$\boldsymbol{\theta}_{t+1} = \text{FO-OPT}(\boldsymbol{\theta}_t, \nabla f(\boldsymbol{\theta}_t)) \tag{2}$$

where $t$ is the iteration number. Ideally, if parallel computing can be used to parallelize the sequential iterations in FOO (i.e., to execute several sequential iterations simultaneously), it will be able to lead to a noticeable improvement in its optimization efficiency since fewer *sequential* iterations will be required for convergence. Unfortunately, there is an inherent iterative dependency in standard FOO, that is, the output of each iteration $t$ (e.g., $\boldsymbol{\theta}_t$) is the input of the next iteration $t+1$. Such an iterative and sequential process makes it nearly impossible to attain $\boldsymbol{\theta}_t$ and $\boldsymbol{\theta}_{t+1}$ concurrently, and therefore parallelize the iterations for established FOO algorithms.

## 4 The OptEx Framework

To this end, we introduce the first general framework in Algo. 1 with a detailed illustration in Fig. 1, namely *first-order optimization expedited with approximately parallelized iterations* (OptEx), to overcome the aforementioned inherent iterative dependency in FOO and facilitate the realization of parallelized iterations therein. To achieve this, we first propose a kernelized gradient estimation with the technique of separable kernel function and local history of gradient to efficiently and effectively estimate the gradient at any input in the domain (Sec. 4.1). We then follow standard FOO algorithms with this kernelized gradient estimation to approximate the inputs for the next $N$ sequential iterations to be parallelized (Sec.4.2), aiming to overcome the inherent iterative dependency in FOO. Lastly, we finish our approximately parallelized iterations by leveraging parallel computing to run standard FOO algorithms on these $N$ inputs concurrently using the ground-truth gradients (Sec. 4.3).

**Algorithm 1:** OptEx

**Input:** FO-OPT, $k(\cdot, \cdot)$, $\boldsymbol{\theta}_0$, $T$, $N$, $\mathcal{G} = \varnothing$

1 **for** *sequential iteration* $t \in [T]$ **do**
2 $\quad$ Initialization: $\boldsymbol{\theta}_{t,0} \leftarrow \boldsymbol{\theta}_{t-1}$
3 $\quad$ Update $\boldsymbol{\mu}_t(\boldsymbol{\theta})$ using (4.1) with $\mathcal{G}$
4 $\quad$ **for** *proxy step* $s \in [N-1]$ **do**
5 $\quad\quad$ $\boldsymbol{\theta}_{t,s} \leftarrow$ FO-OPT$(\boldsymbol{\theta}_{t,s-1}, \boldsymbol{\mu}_t(\boldsymbol{\theta}_{t,s-1}))$
6 $\quad$ **for** *process* $i \in [N]$ *in parallel* **do**
7 $\quad\quad$ Sample $f$ to evaluate $\nabla f(\boldsymbol{\theta}_{t,i-1})$
8 $\quad\quad$ $\boldsymbol{\theta}_t^{(i)} \leftarrow$ FO-OPT$(\boldsymbol{\theta}_{t,i-1}, \nabla f(\boldsymbol{\theta}_{t,i-1}))$
9 $\quad\quad$ $\mathcal{G} \leftarrow \mathcal{G} \cup \{(\boldsymbol{\theta}_{t,i-1}, \nabla f(\boldsymbol{\theta}_{t,i-1}))\}$
10 $\quad$ $\boldsymbol{\theta}_t \leftarrow \boldsymbol{\theta}_t^{(N)}$

Figure 1: An illustration of OptEx at iteration $t$.

## 4.1 Kernelized Gradient Estimation

As mentioned in our Sec. 3, $\nabla F$ is assumed to be sampled from a Gaussian process, i.e., $\nabla F \sim \mathcal{GP}(\mathbf{0}, \mathbf{K}(\cdot, \cdot))$ with kernel function $\mathbf{K}$. Then, for every sequential iteration $t$ of Algo. 1, conditioned on the history of gradients during optimization $\mathcal{G} \triangleq \{(\boldsymbol{\theta}_\tau, \nabla f(\boldsymbol{\theta}_\tau)\}_{\tau=1}^{N(t-1)}$ [1], $\nabla F$ then follows the posterior Gaussian process: $\nabla F \sim \mathcal{GP}\left(\boldsymbol{\mu}_t(\cdot), \boldsymbol{\Sigma}_t^2(\cdot, \cdot)\right)$ with the mean function $\boldsymbol{\mu}_t(\cdot)$ and the covariance function $\boldsymbol{\Sigma}_t^2(\cdot, \cdot)$ defined as below [24]:

$$
\begin{aligned}
\boldsymbol{\mu}_t(\boldsymbol{\theta}) &\triangleq \mathbf{V}_t^\top(\boldsymbol{\theta}) \left(\mathbf{U}_t + \sigma^2 \mathbf{I}\right)^{-1} \text{vec}(\mathbf{G}_t^\top), \\
\boldsymbol{\Sigma}_t^2(\boldsymbol{\theta}, \boldsymbol{\theta}') &\triangleq \mathbf{K}\left(\boldsymbol{\theta}, \boldsymbol{\theta}'\right) - \mathbf{V}_t^\top(\boldsymbol{\theta}) \left(\mathbf{U}_t + \sigma^2 \mathbf{I}\right)^{-1} \mathbf{V}_t\left(\boldsymbol{\theta}'\right)
\end{aligned}
\tag{3}
$$

where $\text{vec}(\cdot)$ vectorizes a matrix into a column vector, $\mathbf{G}_t \triangleq [\nabla f(\boldsymbol{\theta}_\tau)]_{\tau=1}^{N(t-1)}$ is a $d \times N(t-1)$-dimensional matrix, $\mathbf{V}_t^\top(\boldsymbol{\theta}) \triangleq [\mathbf{K}(\boldsymbol{\theta}, \boldsymbol{\theta}_\tau)]_{\tau=1}^{N(t-1)}$ is a $d \times N(t-1)d$-dimensional matrices, and $\mathbf{U}_t \triangleq \left[\mathbf{K}(\boldsymbol{\theta}_\tau, \boldsymbol{\theta}_{\tau'})\right]_{\tau,\tau'=1}^{N(t-1)}$ is a $N(t-1)d \times N(t-1)d$-dimensional matrices. We therefore propose to use $\boldsymbol{\mu}_t(\cdot)$ to estimate the gradient at *any* input $\boldsymbol{\theta} \in \mathbb{R}^d$, that is,

$$
\nabla F(\boldsymbol{\theta}) \approx \mu_t(\boldsymbol{\theta}),
\tag{4}
$$

and covariance $\boldsymbol{\Sigma}^2(\boldsymbol{\theta}) \triangleq \boldsymbol{\Sigma}^2(\boldsymbol{\theta}, \boldsymbol{\theta})$ to measure the quality of this gradient estimation in a principled way, which will be further theoretically supported in our Sec. 5.1.

However, for every sequential iteration $t$ of Algo. 1 with (3), it will incur a computational complexity of $\mathcal{O}(N^3(t-1)^3 d^3)$, along with a space complexity of $\mathcal{O}(N(t-1)d)$. Practically, this presents a significant challenge in scenarios with a large input dimension $d$ or requiring a substantial number $T$ of sequential iterations for convergence, such as in neural network training [8]. To mitigate these complexity issues, we introduce two techniques: the separable kernel function and the local history of gradients, to reduce both the computational and space complexities associated with our kernelized gradient estimation, thereby enhancing its efficiency and practical applicability.

**Separable Kernel Function.** Let $\mathbf{K}(\cdot, \cdot) = k(\cdot, \cdot)\mathbf{I}$ where $k(\cdot, \cdot)$ produces a scalar value and $\mathbf{I}$ is a $d \times d$ identity matrix, and define the $N(t-1)$-dimensional vector $\boldsymbol{k}_t^\top(\boldsymbol{\theta}) \triangleq [k(\boldsymbol{\theta}, \boldsymbol{\theta}_\tau)]_{\tau=1}^{N(t-1)}$, and $N(t-1) \times N(t-1)$-dimensional matrix $\mathbf{K}_t \triangleq [k(\boldsymbol{\theta}_\tau, \boldsymbol{\theta}_{\tau'})]_{\tau=\tau'=1}^{N(t-1)}$, we can prove that the Gaussian process in (3) can be simplified as the Gaussian process in Prop. 4.1 (line 3 of Algo. 1).

**Proposition 4.1.** *Let* $\mathbf{K}(\cdot, \cdot) = k(\cdot, \cdot)\mathbf{I}$, *the posterior mean and covariance in* (3) *become*

$$
\begin{aligned}
\boldsymbol{\mu}_t(\boldsymbol{\theta}) &= \left[\left(\boldsymbol{k}_t^\top(\boldsymbol{\theta}) \left(\mathbf{K}_t + \sigma^2 \mathbf{I}\right)^{-1}\right) \mathbf{G}_t\right]^\top, \\
\boldsymbol{\Sigma}_t^2(\boldsymbol{\theta}, \boldsymbol{\theta}') &= \left(k(\boldsymbol{\theta}, \boldsymbol{\theta}') - \boldsymbol{k}_t^\top(\boldsymbol{\theta}) \left(\mathbf{K}_t + \sigma^2 \mathbf{I}\right)^{-1} \boldsymbol{k}_t(\boldsymbol{\theta}')\right) \mathbf{I}.
\end{aligned}
$$

---

[1] We slightly abuse the notation $f$ to denote the different functions that are randomly sampled per iteration and $(\boldsymbol{\theta}_\tau, \nabla f(\boldsymbol{\theta}_\tau)$ to denote a historical evaluation till sequential iteration $t-1$ with parallelism of $N$.

Its proof is in Appx. A.1. Prop. 4.1 shows that with a separable kernel function $\mathbf{K}(\cdot, \cdot) = k(\cdot, \cdot)\,\mathbf{I}$, the multi-output Gaussian process in a $d$-dimensional space can be effectively decoupled into $d$ independent single-output Gaussian processes. Each of these processes results from the same scalar kernel function $k$, leading to a uniform posterior form shared by all these processes, i.e., the expression $\boldsymbol{k}_t^\top(\boldsymbol{\theta})\left(\mathbf{K}_t + \sigma^2 \mathbf{I}\right)^{-1}$ in $\boldsymbol{\mu}_t(\boldsymbol{\theta})$ and $k(\boldsymbol{\theta}, \boldsymbol{\theta}') - \boldsymbol{k}_t^\top(\boldsymbol{\theta})\left(\mathbf{K}_t + \sigma^2 \mathbf{I}\right)^{-1}\boldsymbol{k}_t(\boldsymbol{\theta}')$ in $\boldsymbol{\Sigma}_t^2(\boldsymbol{\theta}, \boldsymbol{\theta}')$. This thus considerably diminishes the computational complexity, now quantified as $\mathcal{O}(N^3(t-1)^3 + N(t-1)d)$, resulting in a more computationally efficient gradient estimation in practice.

**Local History of Gradients.** Conventional FOO algorithms predominantly operate by optimizing within a localized region neighboring the initial input $\boldsymbol{\theta}_0$ [29]. This therefore indicates that our Algo. 1 only requires precise gradient estimation within a local region. In this context, the use of a local gradient history is posited as sufficiently informative for effective kernelized gradient estimation, which can be supported by the theoretical results in [30] and the empirical evidence in our Sec. 6. As a result, rather than relying on a complete gradient history, we propose to use a localized gradient history of size $T_0$ that neighbors $\boldsymbol{\theta}$ to estimate the gradient at $\boldsymbol{\theta}$. This strategic modification results in a substantial reduction of computational complexity to $\mathcal{O}(T_0^3 + T_0 d)$ as well as a corresponding decrease in space complexity to $\mathcal{O}(T_0 d)$, which is especially beneficial in the situations where $T_0$ is considerably smaller than $N(t-1)$ for $t \in [T]$.

### 4.2 Multi-Step Proxy Updates

The ability of our kernelized gradient estimation to provide gradient estimation at any input $\boldsymbol{\theta}$ then enables the application of a multi-step gradient estimation. This helps to approximate the inputs for the next $N$ sequential iterations $\{\boldsymbol{\theta}_{\tau+i}\}_{i=0}^{N-1}$ to be parallelized in standard FOO, given $\boldsymbol{\theta}\tau$. Specifically, in the context of our Algo. 1, for every sequential iteration $t \in [T]$, by employing a first-order optimizer (FO-OPT), we can approximate the inputs required by our parallelized iteration in Sec. 4.3 *sequentially* as below through our multi-step proxy updates (line 4-5 of Algo. 1).

$$\boldsymbol{\theta}_{t,s} = \text{FO-OPT}(\boldsymbol{\theta}_{t,s-1}, \boldsymbol{\mu}_t(\boldsymbol{\theta}_{t,s-1})), \forall s \in [N-1]. \tag{5}$$

Intuitively, these proxy updates imitate the sequential iterations in standard FOO by using only the estimated gradients in our Sec. 4.1. We will show that these proxy updates can provide a reasonably good approximation of the ground-truth updates in Sec. 5.1. Meanwhile, despite the iterative and sequential nature of (5), our proxy updates based on operations on relatively small-sized matrices (refer to the Prop. 4.1) will still be able to provide significantly enhanced computational efficiency compared to the ground-truth updates based on expensive evaluation of function values and gradients in complex tasks, like neural network training. This effectiveness and efficiency of (5) thus render it an essential foundation for achieving parallelized iterations and improved the optimization efficiency in FOO.

### 4.3 Approximately Parallelized Iterations

Upon obtaining the inputs $\{\boldsymbol{\theta}_{t,s-1}\}_{s=1}^N$ for the next $N$ sequential iterations to be parallelized, we then finish our approximately parallelized iteration by executing standard FOO algorithms over each of $\{\boldsymbol{\theta}_{t,s-1}\}_{s=1}^N$ based on the ground-truth gradients $\{\nabla f(\boldsymbol{\theta}_{t,s-1})\}_{s=1}^N$ *in parallel* (line 6-9 of Algo. 1, see also the processes in Fig. 1). That is,

$$\boldsymbol{\theta}_t^{(i)} = \text{FO-OPT}(\boldsymbol{\theta}_{t,i-1}, \nabla f(\boldsymbol{\theta}_{t,i-1})), \forall i \in [N]. \tag{6}$$

After that, the final input $\boldsymbol{\theta}_t = \boldsymbol{\theta}_t^{(N)}$ will be used in the next sequential iteration (line 10 of Algo. 1). Of note, central to the approximately parallelized iterations in our OptEx framework is the necessity of evaluating the gradients $\{\nabla f(\boldsymbol{\theta}_{t,s-1})\}_{s=1}^N$ in our Algo. 1. These evaluations in fact play pivotal roles in reducing the estimation error of our kernelized gradient estimation and consequently improving the performance of our OptEx by augmenting the gradient history near the input $\boldsymbol{\theta}_t$ with $N$ more evaluations, which will be supported by the theoretical results in our Sec. 5 and the empirical evidence in our Appx. B.3.

## 5 Theoretical Results

To begin with, we formally present the assumptions mentioned in our Sec. 3 as below.

**Assumption 1.** $\nabla f(\boldsymbol{\theta}) - \nabla F(\boldsymbol{\theta})$ follows $\mathcal{N}\left(\mathbf{0}, \sigma^2 \mathbf{I}\right)$ for any $\boldsymbol{\theta} \in \mathbb{R}^d$.

**Assumption 2.** $\nabla F$ is sampled from a Gaussian process $\mathcal{GP}\left(\mathbf{0}, \mathbf{K}(\cdot, \cdot)\right)$ where $\mathbf{K}(\cdot, \cdot) = k(\cdot, \cdot) \mathbf{I}$ and $|k(\boldsymbol{\theta}, \boldsymbol{\theta})| \leq \kappa$ for any $\boldsymbol{\theta} \in \mathbb{R}^d$.

Note that the Assump. 1 has already been widely employed in the literature [21, 22, 23]. Meanwhile, it is also common to assume that $F$ is sampled from a Gaussian process [24, 31], implying that $\nabla F$ follows a Gaussian process as well [24, 25, 26] (Assump. 2), i.e., $\nabla F$ can be any function in this prior. The inclusion of a separable kernel function in Assump. 2 aims to enhance the efficiency of our kernelized gradient estimation in Sec. 4.1 and simplify our theoretical analyses below, whereas our conclusions apply to non-separable kernel functions as well by following our proof techniques.

### 5.1 Gradient Estimation Analysis

Following the principled idea in kernelized bandit [32, 33] and Bayesian Optimization [34, 31], we define the maximal information gain as below

$$\gamma_n \triangleq \max_{\{\boldsymbol{\theta}_j\}_{j=1}^n \subset \mathbb{R}^d} I\left(\text{vec}(\mathbf{G}_n); \text{vec}(\boldsymbol{\nabla}_n)\right) \tag{7}$$

where $I(\text{vec}(\mathbf{G}_n); \text{vec}(\boldsymbol{\nabla}_n))$ is the mutual information between $\mathbf{G}_n \triangleq [\nabla f(\boldsymbol{\theta}_i)]_{i=1}^n$ and $\boldsymbol{\nabla}_n \triangleq [\nabla F(\boldsymbol{\theta}_i)]_{i=1}^n$. In essence, $\gamma_n$ encapsulates the maximum amount of information about $\nabla F$ that can be gleaned from observing any set of $n$ evaluated gradients, represented as $\mathbf{G}_n$, which is known to be problem dependent measure that is highly related to the kernel function $k(\cdot, \cdot)$ [32]. Built on this notation, we then provide the following theoretical result for our gradient estimation.

**Theorem 1** (Gradient Estimation Error). *Let $\delta \in (0, 1)$ and $\alpha \triangleq d + (\sqrt{d} + 1)\ln(1/\delta)$. Given Assump. 1 and 2, let $|\mathcal{G}| = T_0 - 1$ for any sequential iteration $t$ in Algo. 1, then for any $\boldsymbol{\theta} \in \mathbb{R}^d, t > 0$, with a probability of at least $1 - \delta$,*

$$\|\nabla F(\boldsymbol{\theta}) - \boldsymbol{\mu}_t(\boldsymbol{\theta})\| \leq \sqrt{\alpha \left\|\boldsymbol{\Sigma}^2(\boldsymbol{\theta})\right\|} \quad \text{where} \quad \frac{\kappa}{(\kappa + 1/\sigma^2)^{T_0 - 1}} \leq \left\|\boldsymbol{\Sigma}^2(\boldsymbol{\theta})\right\| \leq \frac{4\max\{\kappa, \sigma^2\}\gamma_{T_0}}{T_0 d} .$$

The proof is in Appx. A.2. It is important to note that since FOO pertains to local optimization, the global fulfillment of Assump. 2 is not a prerequisite. That is, the assumption that $\nabla F$ is sampled from $\mathcal{GP}(\mathbf{0}, \mathbf{K})$ within a local region will already be sufficient for our kernelized gradient estimation in Sec. 4.1 to achieve accurate gradient estimation in practice. Our Sec. 6 will later evidence this empirically. Thm. 1 with upper bound on $\left\|\boldsymbol{\Sigma}^2(\boldsymbol{\theta})\right\|$ illustrates that the efficacy of our kernelized gradient estimation in the worst case will enjoy a polynomial error rate of $\mathcal{O}\left(\sqrt{\gamma_{T_0}/T_0}\right)$. This means that if $\gamma_{T_0}/T_0$ will asymptotically approach zero w.r.t. $T_0$, the error of our kernelized gradient estimation method will become significantly small given a large number of evaluated gradients $T_0$. This consequently facilitates the effectiveness of our proxy updates in (5) built on our kernelized gradient estimation to approximate the ground-truth updates when $|\mathcal{G}|$ is sufficiently large. Meanwhile, Thm. 1 with lower bound on $\left\|\boldsymbol{\Sigma}^2(\boldsymbol{\theta})\right\|$ illustrates that our kernelized gradient estimation in the best case may achieve an exponential error rate of $\mathcal{O}\left(\kappa/(\kappa + 1/\sigma^2)^{T_0 - 1}\right)$, which thus further elaborates the efficacy of kernelized gradient estimation in Sec. 4.1 and proxy updates in Sec. 4.2.

It is important to note that the ratio $\gamma_{T_0}/T_0$ has been demonstrated to asymptotically approach zero for a range of kernel functions, as evidenced in existing literature [35]. This therefore underpins the establishment of concrete error bounds for our kernelized gradient estimation where notation $\widetilde{\mathcal{O}}$ is applied to hide the logarithmic factors, delineated as follows:

**Corollary 1** (Concrete Error Bounds). *Let $k(\cdot, \cdot)$ be the radial basis function (RBF) kernel, then*

$$\|\nabla F(\boldsymbol{\theta}) - \boldsymbol{\mu}_t(\boldsymbol{\theta})\| = \widetilde{\mathcal{O}}\left(T_0^{-1/2}\right) .$$

*Let $k(\cdot, \cdot)$ be the Matérn kernel where $\nu$ is the smoothness parameter, then*

$$\|\nabla F(\boldsymbol{\theta}) - \boldsymbol{\mu}_t(\boldsymbol{\theta})\| = \widetilde{\mathcal{O}}\left(T_0^{-\nu/(2\nu + d(d+1))}\right) .$$

Cor. 1 elucidates that with kernel functions such as RBF and Matérn kernel, the error in our kernelized gradient estimation indeed will diminish asymptotically w.r.t. $T_0$. That is, as $T_0$ increases, the estimation error $\|\nabla F(\boldsymbol{\theta}) - \boldsymbol{\mu}_t(\boldsymbol{\theta})\|$ decreases and consequently the proxy updates in (5) become closer to the ground-truth updates. It is important to note that this reduction typically follows a non-linear trajectory, suggesting that the effect of an increasing $T_0$ on our kernelized gradient estimation diminishes when $T_0$ is reasonably large. This consequently affirms the reasonability of our utility of local history for gradient estimation in Sec. 4.1, which leads to not only accurate but also efficient gradient estimations.

## 5.2 Iteration Complexity Analysis

We first introduce Assump. 3, which has been widely applied in stochastic optimization [16, 36], to underpin the analysis of sequential iteration complexity of our OptEx framework.

**Assumption 3.** $F$ is $L$-Lipschitz smooth: $\left\|\nabla F(\boldsymbol{\theta}) - \nabla F(\boldsymbol{\theta}')\right\| \leq L \left\|\boldsymbol{\theta} - \boldsymbol{\theta}'\right\|$ for any $\boldsymbol{\theta}, \boldsymbol{\theta}' \in \mathbb{R}^d$.

To simplify the analysis, we primarily prove the sequential iteration complexity of our SGD-based OptEx where we use $\min_{\tau \in [NT)} \left\|\nabla F(\boldsymbol{\theta}_\tau)\right\|^2$ to denote the minimal gradient norm we can achieve within the whole optimization process when applying our OptEx with $T$ sequential iterations and parallelism of $N$ for a clear and fair comparison with standard SGD. Notably, our analysis can also be extended to other FOO-based OptEx by following similar proof idea.

**Theorem 2** (Upper Bound). *Let* $\delta \in (0,1)$, $\Delta \triangleq F(\boldsymbol{\theta}) - \inf_{\boldsymbol{\theta}} F(\boldsymbol{\theta})$, $\beta \triangleq \max\{\kappa, \sigma^2\}$ *and* $\rho \triangleq (1 - \frac{1}{N})\frac{4\beta\gamma_{T_0}}{\sigma^2 T_0} + \frac{1}{N}$. *Under Assump. 1–3, by choosing* $T \geq \frac{2\Delta L}{N\sigma^2\rho}$, $\eta = \sqrt{\frac{2\Delta}{NLT\sigma^2\rho}}$ *and* $|\mathcal{G}| = T_0 - 1$ *for our SGD-based Algo. 1, with a probability of at least* $1 - \delta$,

$$\min_{t \leq T, s \leq N} \left\|\nabla F(\boldsymbol{\theta}_{t,s})\right\|^2 \leq 2\sigma\sqrt{\frac{2\Delta L\rho}{NT}} + \frac{4\beta \ln(1/2\delta)}{NT} . \tag{8}$$

The proof of Thm. 2 is in Appx. A.3. Of note, our Thm. 2 with $N = 1$ aligns with the established upper bound for standard SGD, as discussed in [36]. Importantly, our Thm. 2 elucidates that with parallelism $N > 1$, our SGD-based OptEx algorithm can expedite the standard SGD by a factor of at least $\sqrt{N/\rho}$, where $1/\rho$ quantifies the impact of the error introduced by our kernelized gradient estimation. This efficiency gain can be further amplified as the accuracy of our kernelized gradient estimation increases (i.e., a decrease in $\rho$), which can be achieved by augmenting the number $T_0$ as discussed in our Sec. 5.1. In addition, Thm. 2 also demonstrates that for a fixed learning rate $\eta$, there exists a constant $N_{\max}$, e.g., $N_{\max} = 2\Delta/(\eta^2 LT\sigma^2\rho)$ in Thm. 2, the parallelism $N$ should roughly remain below to ensure the fastest convergence of function $F$ to a stationary point. In contrast, if $N$ exceeds $N_{\max}$, our SGD-based OptEx will underperform due to the increased gradient estimation error. This observation is further supported by the results presented in Appx. B.3. However, when the learning rate $\eta$ is relatively small (e.g., during fine-tuning in practice), the parallelism $N$ can be significantly larger to achieve a further improved speedup.

**Theorem 3** (Lower Bound). *Let* $\delta \in (0,1)$, $\Delta \triangleq F(\boldsymbol{\theta}) - \inf_{\boldsymbol{\theta}} F(\boldsymbol{\theta})$, $\beta \triangleq \max\{\kappa, \sigma^2\}$, *and* $\widetilde{\beta} \triangleq \min\{\kappa/(\kappa + 1/\sigma^2)^{T_0 - 1}, \sigma^2\}$. *Then, for any* $L > 0, \Delta > 0, N \geq 1, T \geq 1$ *and* $\eta \in [0, 1/L)$, *there exists a* $F$ *on* $\mathbb{R}^d$ $(\forall d > d_0 = \mathcal{O}\left(\beta/(\Delta L^2)\ln NT/\delta\right))$ *satisfying Assump. 1–3 and having the following with a probability of at least* $1 - \delta$ *when applying SGD-based Algo. 1 with* $|\mathcal{G}| = T_0 - 1$,

$$\min_{t \leq T, s \leq N} \left\|\nabla F(\boldsymbol{\theta}_{t,s})\right\|^2 \geq \frac{d_0 \min\{\Delta L, \widetilde{\beta}, 1\}}{4\sqrt{NT}} . \tag{9}$$

The proof of Thm. 3 is in Appx. A.4. Note that when $N = 1$, Thm. 3 aligns with the recognized lower bound for SGD, as elucidated in [37]. Thm. 3 illustrates that, with parallelism of $N$, our SGD-based OptEx can potentially accelerate standard SGD by up to $\sqrt{N}/(\kappa/(\sigma^2(1+1/\sigma^2)^{T_0-1}))$, under the condition that $\kappa/(1+1/\sigma^2)^{T_0-1} \leq \min\{\Delta L, 1, \sigma^2\}$. This upper limit in fact corresponds with the lower bound of the variance in our kernelized gradient estimation, as established in Thm. 1. Essentially, the agreement between Thm. 2 and Thm. 3, in the aspect of parallelism $N$, demonstrates the tightness of our sequential complexity analysis for SGD-based Algo. 1. Finally, the combination of Thm. 2 and Thm. 3 enables us to specify the effective acceleration that can be achieved by our SGD-based OptEx tightly, as shown in our Cor. 2 below.

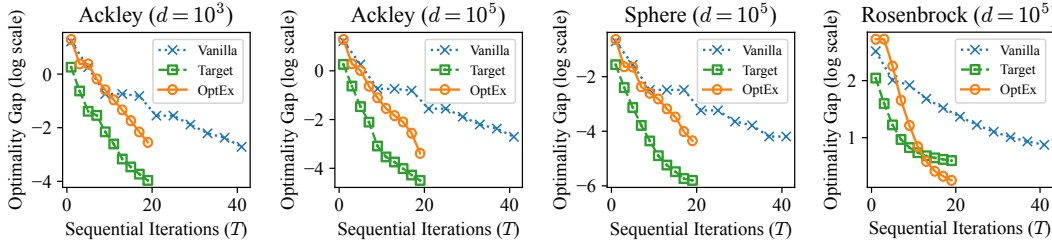

Figure 2: Comparison of the number of sequential iterations $T$ ($x$-axis) required by different methods to achieve the same optimality gap $F(\boldsymbol{\theta}) - \inf_{\boldsymbol{\theta}} F(\boldsymbol{\theta})$ ($y$-axis) for various synthetic functions . The parallelism $N$ is set to 5 and each curve denotes the mean from 5 independent runs.

**Corollary 2** (Acceleration Rate). *With parallelism of $N$, the effective acceleration rate achieved by our SGD-based OptEx over standard SGD is $\Theta(\sqrt{N})$.*

# 6 Experiments

In this section, we use extensive experiments to show that our OptEx framework can considerably enhance the efficiency of FOO with parallel computing, including synthetic experiments (Sec. 6.1), reinforcement learning (Sec. 6.2) and neural network training on various datasets (Sec. 6.3).

## 6.1 Synthetic Function Minimization

Here, we utilize synthetic functions to demonstrate the enhanced performance of our OptEx framework compared to existing baselines, including the standard FOO algorithm, namely `Vanilla`, and FOO with ideally parallelized iterations, namely `Target`, which ideally but impractically utilizes the ground-truth gradient to obtain the inputs for the iterations to be parallelized. More specifically, the `Vanilla` baseline is equivalent to Algo. 1 with parallelism of $N=1$, and the `Target` baseline is equivalent to Algo. 1 with $\boldsymbol{\mu}_t(\boldsymbol{\theta}_{t,s-1})$ being replaced with $\nabla f(\boldsymbol{\theta}_{t,s-1})$, indicating the desired parallelized iteration we aim to approximate. We have also provided a comprehensive illustration of these baselines in Appx. B.1 and detailed experimental setup applied here in Appx. B.2.1.

The results in Fig. 2 with $\sigma^2 = 0$ and $N = 5$ have demonstrated the efficacy of our OptEx framework for deterministic optimization (i.e., $\sigma^2 = 0$). Specifically, Fig. 2 shows that OptEx consistently achieves a notable speedup in optimization efficiency measured by the number of sequential iterations, which is at least $2\times$ more efficient than the `Vanilla` baseline, when optimizing with parallelism of $N = 5$ to reach an equivalent level of optimality gap. This is roughly in line with the result of our Cor. 2, implying the validity of our Cor. 2. Meanwhile, although our OptEx framework slightly underperforms the `Target` baseline, such a phenomenon is in fact quite reasonable since the `Target` baseline can leverage the ground-truth gradient whereas OptEx relies on the kernelized gradient estimation with estimation error bounded in Thm. 1 to parallelize sequential iterations. This also aligns with the insight from our iteration complexity analysis in Thm. 2. Overall, the results in Fig. 2 have provided strong empirical support for the efficacy of our OptEx in expediting FOO, as theoretically justified in our Sec. 5.2. We also present a number of ablation studies as well as analyses in Appx. B.3 to examine the effects of different components in our proposed OptEx framework on its effectiveness.

## 6.2 Reinforcement Learning

We proceed to compare our OptEx framework with previously established baselines under various reinforcement learning tasks with different parameter dimension $d$ from the OpenAI Gym suite [38], with the deployment of DQN agents [39]. Here, the parallelism parameter is set to be $N = 4$ and a detailed experimental setup is provided in Appx. B.2.2. The results are presented in Fig. 3. As illustrated in Fig. 3, the integration of parallel computing techniques, including `Target` and OptEx, considerably outperforms the traditional `Vanilla` baseline in terms of the optimization efficiency quantified by the number of sequential iterations. More importantly, amongst these methodologies, OptEx consistently demonstrates a more stable and superior improvement on the optimization effi-

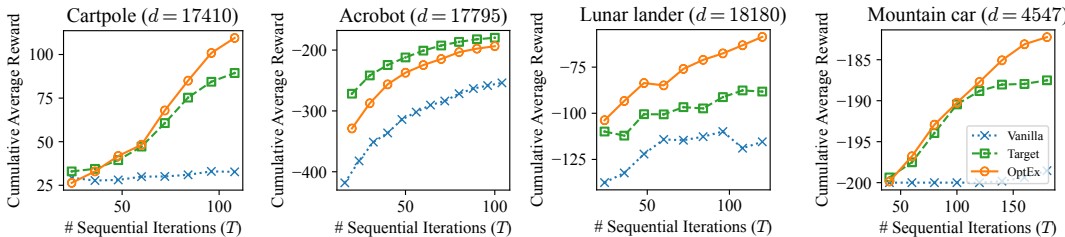

Figure 3: Comparison of the cumulative average reward ($y$-axis) achieved by different methods to train DQN on RL tasks under various parameter dimension $d$ and a varying number of sequential episodes $T$ ($x$-axis). The parallelism $N$ is set to 4 and each curve denotes the mean from 3 independent runs.

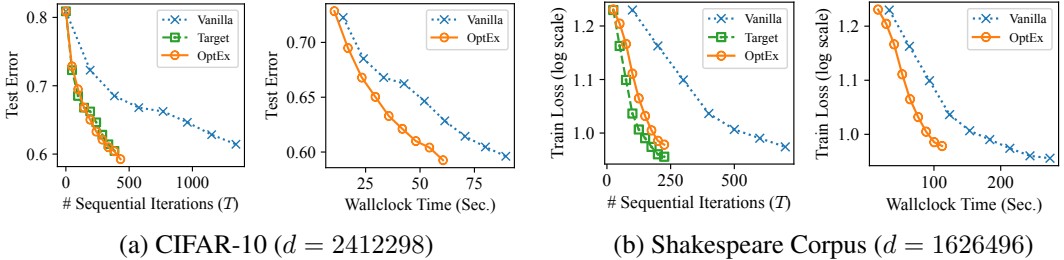

(a) CIFAR-10 ($d = 2412298$)       (b) Shakespeare Corpus ($d = 1626496$)

Figure 4: Comparison of the test error or training loss ($y$-axis) achieved by different optimizers when training deep neural networks on (a) CIFAR-10 and (b) Shakespeare Corpus with a varying number $T$ of sequential iterations or a varying wallclock time ($x$-axis). The parallelism $N$ is set to 4 and each curve denotes the mean from 5 (for CIFAR-10) or 3 (for Shakespeare corpus) independent runs. The wallclock time is evaluated on a single NVIDIA RTX 4090 GPU.

ciency compared with other baselines, which consequently well corroborates the efficacy of OptEx in improving the efficiency of established FOO algorithms. Interestingly, our OptEx framework can even enjoy an improved efficiency over the `Target` baseline where the ground-truth gradient $\nabla f(\cdot)$ is applied. This is likely because the gradient variance (i.e., $\|\mathbf{\Sigma}^2(\boldsymbol{\theta})\|$) in our OptEx framework can asymptotically approach zero by using a large number of history of gradient (refer to our Sec. 4.1), whereas the gradient variance in the `Target` baseline remains the same.

### 6.3 Neural Network Training

At last, we examine the efficacy of our OptEx in expediting the optimization (i.e., training) of deep neural networks, specifically for image classification and text autoregression tasks. Specifically, we apply our OptEx and the aforementioned baselines to (a) train a 10-layer MLP model ($d = 2412298$) with residual connections [40] on CIFAR-10 [41], and (b) train an autoregressive transformer model ($d = 1626496$) borrowed from the Haiku library [42] on a curated collection of works from Shakespeare with parallelism of $N = 4$. Comprehensive details for the experimental setup are provided in Appx. B.2.3 and the final results are illustrated in Fig. 4 where both the number of sequential iterations and wallclock time are used to quantify the optimization efficiency of different optimizers. Intriguingly, as evidenced by Fig. 4, OptEx consistently outperforms `Vanilla` by a large margin in terms of both training and testing errors across the image and text datasets, given an equal number of sequential iterations $T$ or alternatively the same wallclock time budget. Remarkably, the efficiency of OptEx approaches that of the theoretically ideal algorithm – the `Target` baseline, which therefore further verifies the efficacy of our OptEx framework. More results are in Appx. B.3. Overall, these empirical results have well verified the capability of OptEx in significantly expediting FOO algorithms as justified by our theorems in Sec. 5, even in the context of deep neural network training.

## 7 Conclusion

In conclusion, our OptEx framework represents a significant advancement in FOO. By leveraging kernelized gradient estimation to enable approximately parallelized iterations, OptEx effectively re-

duces the number of sequential iterations required for convergence and thus addresses the traditional inefficiencies of FOO. Theoretical analyses and extensive empirical studies validate the reliability and efficacy of OptEx, confirming its potential to expedite optimization processes across various applications. Of note, a limitation of OptEx is the additional storage and computational cost introduced by the kernelized gradient estimation, which we aim to mitigate further in the future work.

## Acknowledgments and Disclosure of Funding

This research is supported by the Guangdong Lab of AI and Digital Economy (SZ) under the Guangming Laboratory Genius Nova Programme (Award No: 24410002).

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

# Appendices

## Appendix A   Proofs

### A.1   Proof of Proposition 4.1

Recall that we have defined $\boldsymbol{k}_t^\top(\boldsymbol{\theta}) \triangleq [k(\boldsymbol{\theta}, \boldsymbol{\theta}_\tau)]_{\tau=1}^{N(t-1)}$, and $\mathbf{K}_t \triangleq [k(\boldsymbol{\theta}_\tau, \boldsymbol{\theta}_{\tau'})]_{\tau=\tau'=1}^{N(t-1)}$. Let $\otimes$ denote the Kronecker product, by introducing the fact that $\mathbf{K}(\cdot, \cdot) = k(\cdot, \cdot)\,\mathbf{I}$ into $\mathbf{V}_t^\top(\boldsymbol{\theta})$ and $\mathbf{U}_t$ from the Gaussian process posterior (3), we have that

$$\mathbf{V}_t^\top(\boldsymbol{\theta}) = [k(\boldsymbol{\theta}, \boldsymbol{\theta}_1)\mathbf{I} \quad \cdots \quad k(\boldsymbol{\theta}, \boldsymbol{\theta}_\tau)\mathbf{I} \quad \cdots \quad k(\boldsymbol{\theta}, \boldsymbol{\theta}_{t-1})\mathbf{I}] = \boldsymbol{k}_t^\top(\boldsymbol{\theta}) \otimes \mathbf{I} \,. \tag{10}$$

Similarly,

$$\mathbf{U}_t = \begin{bmatrix} k(\boldsymbol{\theta}_1, \boldsymbol{\theta}_1)\mathbf{I} & \cdots & k(\boldsymbol{\theta}_1, \boldsymbol{\theta}_{\tau'})\mathbf{I} & \cdots & k(\boldsymbol{\theta}_1, \boldsymbol{\theta}_{t-1})\mathbf{I} \\ \vdots & \vdots & \vdots & \vdots & \vdots \\ k(\boldsymbol{\theta}_\tau, \boldsymbol{\theta}_1)\mathbf{I} & \cdots & k(\boldsymbol{\theta}_\tau, \boldsymbol{\theta}_{\tau'})\mathbf{I} & \cdots & k(\boldsymbol{\theta}_\tau, \boldsymbol{\theta}_{t-1})\mathbf{I} \\ \vdots & \vdots & \vdots & \vdots & \vdots \\ k(\boldsymbol{\theta}_{t-1}, \boldsymbol{\theta}_1)\mathbf{I} & \cdots & k(\boldsymbol{\theta}_{t-1}, \boldsymbol{\theta}_{\tau'})\mathbf{I} & \cdots & k(\boldsymbol{\theta}_{t-1}, \boldsymbol{\theta}_{t-1})\mathbf{I} \end{bmatrix} = \mathbf{K}_t \otimes \mathbf{I} \,. \tag{11}$$

By introducing the results above into the posterior mean and variance in (3), we have

$$\begin{aligned} \boldsymbol{\mu}_t(\boldsymbol{\theta}) &\overset{(a)}{=} \mathbf{V}_t^\top(\boldsymbol{\theta}) \left(\mathbf{U}_t + \sigma^2 \mathbf{I}\right)^{-1} \mathrm{vec}(\mathbf{G}_t^\top) \\[2mm] &\overset{(b)}{=} \left(\boldsymbol{k}_t^\top(\boldsymbol{\theta}) \otimes \mathbf{I}\right) \left(\mathbf{K}_t \otimes \mathbf{I} + \sigma^2 \mathbf{I}\right)^{-1} \mathrm{vec}(\mathbf{G}_t^\top) \\[2mm] &\overset{(c)}{=} \left(\boldsymbol{k}_t^\top(\boldsymbol{\theta}) \otimes \mathbf{I}\right) \left[\left(\mathbf{K}_t + \sigma^2 \mathbf{I}\right) \otimes \mathbf{I}\right]^{-1} \mathrm{vec}(\mathbf{G}_t^\top) \\[2mm] &\overset{(d)}{=} \left(\boldsymbol{k}_t^\top(\boldsymbol{\theta}) \otimes \mathbf{I}\right) \left[\left(\mathbf{K}_t + \sigma^2 \mathbf{I}\right)^{-1} \otimes \mathbf{I}\right] \mathrm{vec}(\mathbf{G}_t^\top) \\[2mm] &\overset{(e)}{=} \left(\left[\boldsymbol{k}_t^\top(\boldsymbol{\theta}) \left(\mathbf{K}_t + \sigma^2 \mathbf{I}\right)^{-1}\right] \otimes \mathbf{I}\right) \mathrm{vec}\left(\mathbf{G}_t^\top\right) \\[2mm] &\overset{(f)}{=} \mathrm{vec}\left(\mathbf{G}_t^\top \left[\boldsymbol{k}_t^\top(\boldsymbol{\theta}) \left(\mathbf{K}_t + \sigma^2 \mathbf{I}\right)^{-1}\right]^\top\right) \\[2mm] &\overset{(g)}{=} \left[\left(\boldsymbol{k}_t^\top(\boldsymbol{\theta}) \left(\mathbf{K}_t + \sigma^2 \mathbf{I}\right)^{-1}\right) \mathbf{G}_t\right]^\top \end{aligned} \tag{12}$$

where $(c)$ come from the bi-linearity of the Kronecker product, i.e., $(\mathbf{A}+\mathbf{B})\otimes\mathbf{C} = \mathbf{A}\otimes\mathbf{C}+\mathbf{B}\otimes\mathbf{C}$ while $(d)$ is from the inverse of the Kronecker product, i.e., $(\mathbf{A} \otimes \mathbf{B})^{-1} = \mathbf{A}^{-1} \otimes \mathbf{B}^{-1}$. In addition, $(e)$ is due to the mixed-product property of the Kronecker product, i.e., $(\mathbf{A}\otimes\mathbf{B})(\mathbf{C}\otimes\mathbf{D}) = (\mathbf{AC})\otimes(\mathbf{BD})$, and $(f)$ results from the mixed Kronecker matrix-vector product of the Kronecker product, i.e., $(\mathbf{A} \otimes \mathbf{B})\mathrm{vec}(\mathbf{C}) = \mathrm{vec}(\mathbf{BCA}^\top)$.

Similarly,

$$\begin{aligned} \boldsymbol{\Sigma}_t^2(\boldsymbol{\theta}, \boldsymbol{\theta}') &\overset{(a)}{=} \mathbf{K}(\boldsymbol{\theta}, \boldsymbol{\theta}') - \mathbf{V}_t^\top(\boldsymbol{\theta}) \left(\mathbf{U}_t + \sigma^2 \mathbf{I}\right)^{-1} \boldsymbol{\Phi}_n(\boldsymbol{\theta}') \\[2mm] &\overset{(b)}{=} k(\boldsymbol{\theta}, \boldsymbol{\theta}')\mathbf{I} - \left(\left[\boldsymbol{k}_t^\top(\boldsymbol{\theta}) \left(\mathbf{K}_t + \sigma^2 \mathbf{I}\right)^{-1}\right] \otimes \mathbf{I}\right) (\boldsymbol{k}_t(\boldsymbol{\theta}') \otimes \mathbf{I}) \\[2mm] &\overset{(c)}{=} k(\boldsymbol{\theta}, \boldsymbol{\theta}')\mathbf{I} - \left(\boldsymbol{k}_t^\top(\boldsymbol{\theta}) \left(\mathbf{K}_t + \sigma^2 \mathbf{I}\right)^{-1} \boldsymbol{k}_t(\boldsymbol{\theta}')\right) \mathbf{I} \\[2mm] &\overset{(d)}{=} \left(k(\boldsymbol{\theta}, \boldsymbol{\theta}') - \boldsymbol{k}_t^\top(\boldsymbol{\theta}) \left(\mathbf{K}_t + \sigma^2 \mathbf{I}\right)^{-1} \boldsymbol{k}_t(\boldsymbol{\theta}')\right) \mathbf{I} \end{aligned} \tag{13}$$

where $(b)$ comes from the result in (12), $(c)$ results from the mixed-product property of the Kronecker product and the fact that $\left( k_t^\top(\boldsymbol{\theta}) \left( \mathbf{K}_t + \sigma^2 \mathbf{I} \right)^{-1} k_t(\boldsymbol{\theta}') \right)$ is a scalar. This finally concludes our proof.

## A.2 Proof of Theorem 1

To begin with, we introduce the following lemmas:

**Lemma A.1** ([43]). *Let* $\boldsymbol{\zeta} \sim \mathcal{N}(\mathbf{0}, \mathbf{I}_d)$ *and* $\delta \in (0, 1)$ *then*

$$\mathbb{P} \left( \|\boldsymbol{\zeta}\|_2 \le \sqrt{d + 2(\sqrt{d} + 1)\ln(1/\delta)} \right) \ge 1 - \delta . \tag{14}$$

**Lemma A.2** (Lemma 2 in Appx. B of [34]). *For any* $\sigma \in \mathbb{R}$ *and any matrix* $\mathbf{A}$, *the following hold*

$$\mathbf{I} - \mathbf{A}^\top \left( \mathbf{A}\mathbf{A}^\top + \sigma^2 \mathbf{I} \right)^{-1} \mathbf{A} = \sigma^2 \left( \mathbf{A}^\top \mathbf{A} + \sigma^2 \mathbf{I} \right)^{-1} . \tag{15}$$

**Lemma A.3** (Sherman-Morrison formula). *For any invertible square matrix* $\mathbf{A}$ *and column vectors* $\boldsymbol{u}, \boldsymbol{v}$, *suppose* $\mathbf{A} + \boldsymbol{u}\boldsymbol{v}^\top$ *is invertible, then the following holds*

$$\left( \mathbf{A} + \boldsymbol{u}\boldsymbol{v}^\top \right)^{-1} = \mathbf{A}^{-1} - \frac{\mathbf{A}^{-1}\boldsymbol{u}\boldsymbol{v}^\top \mathbf{A}^{-1}}{1 + \boldsymbol{v}^\top \mathbf{A}^{-1}\boldsymbol{u}} . \tag{16}$$

**Lemma A.4** (Non-Increasing Variance Norm). *Define variance* $\boldsymbol{\Sigma}_n^2(\boldsymbol{\theta}) \triangleq \boldsymbol{\Sigma}_n^2(\boldsymbol{\theta}, \boldsymbol{\theta})$ *with* $n$ *being the number of gradients employed to evaluate the mean and covariance in Prop. 4.1. Then for any* $\boldsymbol{\theta} \in \mathbb{R}^d$ *and* $n \ge 1$,

$$\left\| \boldsymbol{\Sigma}_n^2(\boldsymbol{\theta}) \right\| \le \left\| \boldsymbol{\Sigma}_{n-1}^2(\boldsymbol{\theta}) \right\| . \tag{17}$$

*Proof.* We follow the idea in [34] and [25] to prove it. Specifically, we firstly define $k(\boldsymbol{\theta}, \boldsymbol{\theta}') = \phi(\boldsymbol{\theta})^\top \phi(\boldsymbol{\theta}')$ and $\boldsymbol{\phi}_n \triangleq [\phi(\boldsymbol{\theta}_i)]_{i=1}^n$. Then the matrix $\mathbf{K}_n$ in Prop. 4.1 can be reformulated as

$$\mathbf{K}_n = \boldsymbol{\phi}_n^\top \boldsymbol{\phi}_n , \tag{18}$$

and based on the definition of $\boldsymbol{\Phi}_n \triangleq \boldsymbol{\phi}_n \boldsymbol{\phi}_n^\top + \sigma^2 \mathbf{I}$,

$$\begin{aligned} \boldsymbol{\Sigma}_t^2(\boldsymbol{\theta}) &\overset{(a)}{=} \left( \phi(\boldsymbol{\theta})^\top \phi(\boldsymbol{\theta}) - \phi(\boldsymbol{\theta})^\top \boldsymbol{\phi}_n \left( \boldsymbol{\phi}_n^\top \boldsymbol{\phi}_n + \sigma^2 \mathbf{I} \right)^{-1} \boldsymbol{\phi}_n^\top \phi(\boldsymbol{\theta}) \right) \mathbf{I} \\ &\overset{(b)}{=} \left( \phi(\boldsymbol{\theta})^\top \left( \mathbf{I} - \boldsymbol{\phi}_n \left( \boldsymbol{\phi}_n^\top \boldsymbol{\phi}_n + \sigma^2 \mathbf{I} \right)^{-1} \boldsymbol{\phi}_n^\top \right) \phi(\boldsymbol{\theta}) \right) \mathbf{I} \\ &\overset{(c)}{=} \left( \sigma^2 \phi(\boldsymbol{\theta})^\top \left( \boldsymbol{\phi}_n \boldsymbol{\phi}_n^\top + \sigma^2 \mathbf{I} \right)^{-1} \phi(\boldsymbol{\theta}) \right) \mathbf{I} \\ &\overset{(d)}{=} \left( \sigma^2 \phi(\boldsymbol{\theta})^\top \boldsymbol{\Phi}_n^{-1} \phi(\boldsymbol{\theta}) \right) \mathbf{I} \end{aligned} \tag{19}$$

where $(c)$ comes from Lemma A.2 by replacing the matrix $\mathbf{A}$ in Lemma A.2 with the matrix $\boldsymbol{\phi}_n^\top$.

As a result,

$$\boldsymbol{\Sigma}_n^2(\boldsymbol{\theta})$$

$$\overset{(a)}{=} \left( \sigma^2 \phi(\boldsymbol{\theta})^\top \boldsymbol{\Phi}_n^{-1} \phi(\boldsymbol{\theta}) \right) \mathbf{I}$$

$$\overset{(b)}{=} \left( \sigma^2 \phi(\boldsymbol{\theta})^\top \left( \boldsymbol{\phi}_{n-1} \boldsymbol{\phi}_{n-1}^\top + \sigma^2 \mathbf{I} + \phi(\boldsymbol{\theta}_n)\phi(\boldsymbol{\theta}_n)^\top \right)^{-1} \phi(\boldsymbol{\theta}) \right) \mathbf{I}$$

$$\overset{(c)}{=} \left( \sigma^2 \phi(\boldsymbol{\theta})^\top \left( \boldsymbol{\Phi}_{n-1} + \phi(\boldsymbol{\theta}_n)\phi(\boldsymbol{\theta}_n)^\top \right)^{-1} \phi(\boldsymbol{\theta}) \right) \mathbf{I}$$

$$\overset{(d)}{=} \left( \sigma^2 \phi(\boldsymbol{\theta})^\top \boldsymbol{\Phi}_{n-1}^{-1} \phi(\boldsymbol{\theta}) - \sigma^2 \left( 1 + \phi(\boldsymbol{\theta}_n)^\top \boldsymbol{\Phi}_{n-1}^{-1} \phi(\boldsymbol{\theta}_n) \right)^{-1} \phi(\boldsymbol{\theta})^\top \boldsymbol{\Phi}_{n-1}^{-1} \phi(\boldsymbol{\theta}_n)\phi(\boldsymbol{\theta}_n)^\top \boldsymbol{\Phi}_{n-1}^{-1} \phi(\boldsymbol{\theta}) \right) \mathbf{I}$$

$$\overset{(e)}{=} \boldsymbol{\Sigma}_{n-1}^2(\boldsymbol{\theta}) - \sigma^2 \left( 1 + \phi(\boldsymbol{\theta}_n)^\top \boldsymbol{\Phi}_{n-1}^{-1} \phi(\boldsymbol{\theta}_n) \right)^{-1} \phi(\boldsymbol{\theta})^\top \boldsymbol{\Phi}_{n-1}^{-1} \phi(\boldsymbol{\theta}_n)\phi(\boldsymbol{\theta}_n)^\top \boldsymbol{\Phi}_{n-1}^{-1} \phi(\boldsymbol{\theta}) \mathbf{I}$$

$$\overset{(f)}{\preccurlyeq} \boldsymbol{\Sigma}_{n-1}^2(\boldsymbol{\theta})$$

(20)

where $(b)$ is due to the fact that $\boldsymbol{\Phi}_n \boldsymbol{\Phi}_n^\top = \boldsymbol{\Phi}_{n-1} \boldsymbol{\Phi}_{n-1}^\top + \phi(\boldsymbol{\theta}_n)\phi(\boldsymbol{\theta}_n)^\top$, and $(d)$ is from Lemma A.3. Finally, $(f)$ derives from the positive semi-definite property of $\boldsymbol{\Phi}_{n-1}^{-1}\phi(\boldsymbol{\theta}_t)\phi(\boldsymbol{\theta}_t)^\top \boldsymbol{\Phi}_{n-1}^{-1}$ and $\boldsymbol{\Phi}_{n-1}^{-1}$, leading to the conclusion of our proof. $\square$

**Lemma A.5** (lower Bound of Variance Norm). *Following the definition in Lemma A.4, for any* $\boldsymbol{\theta} \in \mathbb{R}^d$ *and* $n \geq 1$,

$$\left\| \boldsymbol{\Sigma}_n^2(\boldsymbol{\theta}) \right\| \geq \frac{1}{(\kappa + 1/\sigma^2)} \left\| \boldsymbol{\Sigma}_{n-1}^2(\boldsymbol{\theta}) \right\| . \tag{21}$$

*Proof.* Again, we follow the idea in [34] and [25] to prove it. we first prove the following inequality

$$\left\| \boldsymbol{\Phi}_{n-1}^{-1/2} \phi(\boldsymbol{\theta}_n)\phi(\boldsymbol{\theta}_n)^\top \boldsymbol{\Phi}_{n-1}^{-1/2} \right\| \overset{(a)}{=} \left\| \phi(\boldsymbol{\theta}_n)^\top \boldsymbol{\Phi}_{n-1}^{-1/2} \right\|^2$$

$$\overset{(b)}{=} \phi(\boldsymbol{\theta}_n)^\top \boldsymbol{\Phi}_{n-1}^{-1} \phi(\boldsymbol{\theta}_n)$$

$$\overset{(c)}{\leq} \phi(\boldsymbol{\theta}_n)^\top \boldsymbol{\Phi}_{n-2}^{-1} \phi(\boldsymbol{\theta}_n) \tag{22}$$

$$\overset{(d)}{\leq} \sigma^2 \phi(\boldsymbol{\theta}_n)^\top \phi(\boldsymbol{\theta}_n)$$

$$\overset{(e)}{\leq} \kappa \sigma^2$$

where $(c)$ comes from the fact that $\boldsymbol{\Phi}_{n-1} = \boldsymbol{\Phi}_{n-2}^{-1} + \phi(\boldsymbol{\theta}_{n-1}\phi(\boldsymbol{\theta}_{n-1})^\top \succcurlyeq \boldsymbol{\Phi}_{n-2} \succcurlyeq \cdots \succcurlyeq \sigma^2 \mathbf{I}$ and $(e)$ is due to the fact that $\phi(\boldsymbol{\theta}_n)^\top \phi(\boldsymbol{\theta}_n) = k(\boldsymbol{\theta}_n, \boldsymbol{\theta}_n) \leq \kappa$.

Then, based on the reformulation of $\boldsymbol{\Sigma}_n^2(\boldsymbol{\theta})$ in (20), we have that

$$\boldsymbol{\Sigma}_n^2(\boldsymbol{\theta}) \overset{(a)}{=} \left( \sigma^2 \phi(\boldsymbol{\theta})^\top \left( \boldsymbol{\Phi}_{n-1} + \phi(\boldsymbol{\theta}_n)\phi(\boldsymbol{\theta}_n)^\top \right)^{-1} \phi(\boldsymbol{\theta}) \right) \mathbf{I}$$

$$\overset{(b)}{=} \left( \sigma^2 \phi(\boldsymbol{\theta})^\top \boldsymbol{\Phi}_{n-1}^{-1/2} \left( \mathbf{I} + \boldsymbol{\Phi}_{n-1}^{-1/2} \phi(\boldsymbol{\theta}_n)\phi(\boldsymbol{\theta}_n)^\top \boldsymbol{\Phi}_{n-1}^{-1/2} \right)^{-1} \boldsymbol{\Phi}_{n-1}^{-1/2} \phi(\boldsymbol{\theta}) \right) \mathbf{I}$$

$$\overset{(c)}{\succcurlyeq} \frac{\sigma^2}{1 + \kappa \sigma^2} \phi(\boldsymbol{\theta})^\top \boldsymbol{\Phi}_{n-1}^{-1} \phi(\boldsymbol{\theta}) \mathbf{I}$$

$$\overset{(d)}{=} \frac{\sigma^2}{1 + \kappa \sigma^2} \boldsymbol{\Sigma}_{n-1}^2(\boldsymbol{\theta})$$

(23)

where $(c)$ comes from (22). This finally concludes our proof. $\square$

**Lemma A.6** (Information Gain). *Define* $\mathbf{G}_n \triangleq [\nabla f(\boldsymbol{\theta}_i)]_{i=1}^n$, $\boldsymbol{\nabla}_n \triangleq [\nabla F(\boldsymbol{\theta}_i)]_{i=1}^n$, *and* $\mathbf{K}_n \triangleq [k(\boldsymbol{\theta}_i, \boldsymbol{\theta}_j)]_{i,j=1}^n$. *The information gain* $I(\text{vec}(\mathbf{G}_n); \text{vec}(\boldsymbol{\nabla}_n))$ *has the following form with Assump. 1, 2:*

$$I(\text{vec}(\mathbf{G}_n); \text{vec}(\boldsymbol{\nabla}_n)) = \frac{d}{2} \ln \left( \det(\mathbf{I} + \sigma^{-2}\mathbf{K}_n) \right) . \tag{24}$$

*Proof.* Based on our Assump. 1, 2, the following holds respectively:

$$\text{vec}(\mathbf{G}_n) \mid \text{vec}(\boldsymbol{\nabla}_n) \sim \mathcal{N}(\mathbf{0}, \sigma^2 \mathbf{I}_{nd}), \text{ and } \text{vec}(\mathbf{G}_n) \sim \mathcal{GP}\left(\mathbf{0}, \left(\mathbf{K}_n + \sigma^2 \mathbf{I}_n\right) \otimes \mathbf{I}_d\right) . \tag{25}$$

Due to the fact that $H(\mathbf{z}) = \frac{1}{2} \ln(\det(2\pi e \boldsymbol{\Sigma}))$ if $\mathbf{z} \sim \mathcal{N}(\mu, \boldsymbol{\Sigma})$, the following holds

$$
\begin{aligned}
I(\text{vec}(\mathbf{G}_n); \text{vec}(\boldsymbol{\nabla}_n)) &\overset{(a)}{=} H(\text{vec}(\mathbf{G}_n)) - H(\text{vec}(\mathbf{G}_n) \mid \text{vec}(\boldsymbol{\nabla}_n)) \\
&\overset{(b)}{=} \frac{1}{2} \ln \left( \det \left( 2\pi e \left( \mathbf{K}_n + \sigma^2 \mathbf{I}_n \right) \otimes \mathbf{I}_d \right) \right) - \frac{1}{2} \ln \left( \det \left( 2\pi e \sigma^2 \mathbf{I}_{nd} \right) \right) \\
&\overset{(c)}{=} \frac{1}{2} \ln \left( \left[ \det \left( 2\pi e \left( \mathbf{K}_n + \sigma^2 \mathbf{I}_n \right) \right) \right]^d (\det(\mathbf{I}_d))^n \right) - \frac{1}{2} \ln \left( \det \left( 2\pi e \sigma^2 \mathbf{I}_{nd} \right) \right) \\
&\overset{(d)}{=} \frac{1}{2} \ln \left( \frac{\det(2\pi e (\mathbf{K}_n + \sigma^2 \mathbf{I}_n))}{\det(2\pi e \sigma^2 \mathbf{I}_n)} \right)^d \\
&\overset{(e)}{=} \frac{d}{2} \ln \left( \det(\mathbf{I} + \sigma^{-2}\mathbf{K}_n) \right)
\end{aligned}
$$
$$\tag{26}$$

where $(a)$ comes from the definition of information gain, $(b)$ derives from the results in (25), and $(c)$ is due to the fact that $\det(\mathbf{A} \otimes \mathbf{B}) = (\det(\mathbf{A}))^b (\det(\mathbf{B}))^a$ given the $a \times a$-dimensional matrix $\mathbf{A}$ and $b \times b$-dimensional matrix $\mathbf{B}$. In addition, $(e)$ follows from $\det(\mathbf{A}\mathbf{B}^{-1}) = \det(\mathbf{A})/\det(\mathbf{B})$. This then concludes our proof. $\qquad\square$

**Lemma A.7** (Sum of Variance). *Define the maximal information gain*

$$\gamma_n \triangleq \max_{\{\boldsymbol{\theta}_j\}_{j=1}^n \subset \mathbb{R}^d} I(\text{vec}(\mathbf{G}_n); \text{vec}(\boldsymbol{\nabla}_n)) , \tag{27}$$

*the following then holds*

$$\frac{1}{n} \sum_{i=0}^{n-1} \left\| \boldsymbol{\Sigma}_i^2(\boldsymbol{\theta}) \right\| \leq \frac{2\sigma^2 \gamma_n}{d} . \tag{28}$$

*Proof.* To begin with, we show the following inequalities resulting from the matrix determinant lemma:

$$
\begin{aligned}
\det(\boldsymbol{\Phi}_{i+1}) &= \det \left( \boldsymbol{\Phi}_i + \phi(\boldsymbol{\theta})\phi(\boldsymbol{\theta})^\top \right) \\
&= \det(\boldsymbol{\Phi}_i) \left( 1 + \phi(\boldsymbol{\theta})^\top \boldsymbol{\Phi}_i^{-1} \phi(\boldsymbol{\theta}) \right) .
\end{aligned}
$$
$$\tag{29}$$

Given $\kappa \leq \sigma^2$, since $\left\|\mathbf{\Sigma}_n^2(\boldsymbol{\theta})\right\| \leq \left\|\mathbf{\Sigma}_{n-1}^2(\boldsymbol{\theta})\right\| \leq \cdots \leq \left\|\mathbf{\Sigma}_0^2(\boldsymbol{\theta})\right\| = |k(\boldsymbol{\theta}, \boldsymbol{\theta})| \leq \kappa$ from Lemma A.4, we then have $\phi(\boldsymbol{\theta})^\top \mathbf{\Phi}_i^{-1}\phi(\boldsymbol{\theta}) \leq 1$. As a result,

$$
\begin{aligned}
\frac{1}{2}\sum_{i=0}^{n-1}\left\|\mathbf{\Sigma}_i^2(\boldsymbol{\theta})\right\| &\stackrel{(a)}{=} \sum_{i=0}^{n-1}\frac{1}{2}\sigma^2\phi(\boldsymbol{\theta})^\top \mathbf{\Phi}_i^{-1}\phi(\boldsymbol{\theta}) \\
&\stackrel{(b)}{\leq} \sum_{i=0}^{n-1}\sigma^2 \ln\left(1 + \phi(\boldsymbol{\theta})^\top \mathbf{\Phi}_i^{-1}\phi(\boldsymbol{\theta})\right) \\
&\stackrel{(c)}{=} \sigma^2 \sum_{i=0}^{n-1} \ln\left(\frac{\det(\mathbf{\Phi}_{i+1})}{\det(\mathbf{\Phi}_i)}\right) \\
&\stackrel{(d)}{=} \sigma^2 \ln\left(\frac{\det(\mathbf{\Phi}_n)}{\det(\mathbf{\Phi}_0)}\right) \\
&\stackrel{(e)}{=} \sigma^2 \ln\left(\frac{\det(\phi_n \phi_n^\top + \sigma^2 \mathbf{I})}{\det(\sigma^2 \mathbf{I})}\right) \\
&\stackrel{(f)}{=} \sigma^2 \ln\left(\det(\sigma^{-2}\phi_n\phi_n^\top + \mathbf{I})\right) \\
&\stackrel{(g)}{=} \sigma^2 \ln\left(\det(\mathbf{I} + \sigma^{-2}\phi_n^\top \phi_n)\right) \\
&\stackrel{(h)}{\leq} \frac{2\sigma^2 \gamma_n}{d}
\end{aligned}
\tag{30}
$$

where $(a)$ follows from the reformulation of $\mathbf{\Sigma}_i^2(\boldsymbol{\theta})$ in (19), $(b)$ results from the fact that $x/2 \leq \ln(1+x)$ for any $x \in (0, 1)$, $(c)$ derives from (29), $(d)$ is from the telescoping sum, $(e)$ is due to the fact that $\det(\mathbf{\Phi}_0) = \det(\sigma^2 \mathbf{I})$, $(f)$ is from the fact that $\det(\mathbf{A}\mathbf{B}^{-1}) = \det(\mathbf{A})/\det(\mathbf{B})$, $(g)$ comes from the Sylvester's determinant identity, i.e., $\det(\mathbf{\Phi}_i) = \det(\mathbf{K}_i + \sigma^2 \mathbf{I}_i)$ according to the definition of $\mathbf{\Phi}_i$, and $(h)$ results from the fact that $\mathbf{K}_n = \phi_n^\top \phi_n$ in (18), the conclusion in Lemma A.6, and the definition of $\gamma_n$.

Following the same idea, given $\kappa > \sigma^2$, we have

$$
\begin{aligned}
\frac{1}{2\kappa}\sum_{i=0}^{n-1}\left\|\mathbf{\Sigma}_i^2(\boldsymbol{\theta})\right\| &\stackrel{(a)}{=} \sum_{i=0}^{n-1}\frac{\sigma^2}{2\kappa}\phi(\boldsymbol{\theta})^\top \mathbf{\Phi}_i^{-1}\phi(\boldsymbol{\theta}) \\
&\stackrel{(b)}{\leq} \sum_{i=0}^{n-1} \ln\left(1 + \frac{\sigma^2}{\kappa}\phi(\boldsymbol{\theta})^\top \mathbf{\Phi}_i^{-1}\phi(\boldsymbol{\theta})\right) \\
&\stackrel{(c)}{\leq} \sum_{i=0}^{n-1} \ln\left(1 + \phi(\boldsymbol{\theta})^\top \mathbf{\Phi}_i^{-1}\phi(\boldsymbol{\theta})\right) \\
&\stackrel{(d)}{\leq} \frac{2\gamma_n}{d} \; .
\end{aligned}
\tag{31}
$$

Combining the results in (30) and (31), we conclude our proof by

$$
\frac{1}{n}\sum_{i=0}^{n-1}\left\|\mathbf{\Sigma}_i^2(\boldsymbol{\theta})\right\| \leq \frac{4\max\{\kappa, \sigma^2\}\gamma_n}{dn} \; .
\tag{32}
$$

$\square$

*Proof of our Thm. 1.* Since $\mathbf{\Sigma}_n^{-1}(\boldsymbol{\theta})\left[\boldsymbol{\mu}_n(\boldsymbol{\theta}) - \nabla F(\boldsymbol{\theta})\right] \sim \mathcal{N}(\mathbf{0}, \mathbf{I}_d)$, according to Lemma A.1, for any $\delta \in (0, 1)$ and $\alpha \triangleq d + 2(\sqrt{d} + 1)\ln(1/\delta)$, with a probability of at least $1 - \delta$,

$$
\begin{aligned}
\|\nabla F(\boldsymbol{\theta}) - \boldsymbol{\mu}_n(\boldsymbol{\theta})\| &\stackrel{(a)}{\leq} \|\mathbf{\Sigma}_n(\boldsymbol{\theta})\| \left\|\mathbf{\Sigma}_n^{-1}(\boldsymbol{\theta})\left[\boldsymbol{\mu}_n(\boldsymbol{\theta}) - \nabla F(\boldsymbol{\theta})\right]\right\| \\
&\stackrel{(b)}{\leq} \sqrt{\alpha}\left\|\mathbf{\Sigma}_n^2(\boldsymbol{\theta})\right\|^{1/2}
\end{aligned}
\tag{33}
$$

where $(a)$ is from CauchySchwarz inequality and $(b)$ is from Lemma A.1. By introducing the results in Lemma A.5 and Lemma A.7 into the result above and letting $T_0 = n + 1$, we conclude our proof.

### A.3 Proof of Theorem 2

In general, we follow the idea in [36] to give a high probability convergence for our OptEx algorithm. To begin with, we introduce the following definition and lemma.

**Definition A.1** (Sub-Gaussian Random Variable). A random variable X is $\sigma$-sub-Gaussian if $\mathbb{E}\left[\exp\left(\lambda^2 X^2\right)\right] \leq \exp\left(\lambda^2 \sigma^2\right) \forall \lambda$ such that $|\lambda| \leq \frac{1}{\sigma}$.

**Lemma A.8** (Bound for Sub-Gaussian Random Variable). *Suppose X is a $\sigma$-sub-Gaussian random variable, then for any $a \in \mathbb{R}, 0 \leq b \leq \frac{1}{2\sigma}$,*

$$\mathbb{E}\left[\exp\left(aX + b^2 X^2\right)\right] \leq \exp\left((a^2 + b^2)\sigma^2 + \frac{1}{4}\right) . \tag{34}$$

*Proof.*

$$
\begin{aligned}
\mathbb{E}\left[\exp\left(aX + b^2 X^2\right)\right] &\overset{(a)}{\leq} \mathbb{E}\left[\exp\left(a^2\sigma^2 + \frac{X^2}{4\sigma^2} + b^2 X^2\right)\right] \\
&\overset{(b)}{=} \exp\left(a^2\sigma^2\right) \mathbb{E}\left[\exp\left(\left(\frac{1}{4\sigma^2} + b^2\right) X^2\right)\right] \\
&\overset{(c)}{\leq} \exp\left(a^2\sigma^2\right) \exp\left(\left(\frac{1}{4\sigma^2} + b^2\right)\sigma^2\right) \\
&\overset{(d)}{=} \exp\left((a^2 + b^2)\sigma^2 + \frac{1}{4}\right)
\end{aligned}
\tag{35}
$$

where $(c)$ comes from the definition of $\sigma$-sub-Gaussian random variable. $\square$

*Proof of Thm. 2.* Define

$$
\begin{aligned}
\sigma^2(\boldsymbol{\theta}_{t,s}) &\triangleq \begin{cases} \left\|\boldsymbol{\Sigma}^2(\boldsymbol{\theta}_{t,s}, \boldsymbol{\theta}_{t,s})\right\| & \text{if } s < N - 1 \\ \sigma^2 & \text{if } s = N - 1 \end{cases}, \\
\boldsymbol{\varepsilon}(\boldsymbol{\theta}_{t,s}) &\triangleq \begin{cases} \nabla F(\boldsymbol{\theta}_{t,s}) - \nabla\boldsymbol{\mu}_t(\boldsymbol{\theta}_{t,s}) & \text{if } s < N - 1 \\ \nabla F(\boldsymbol{\theta}_{t,s}) - \nabla f(\boldsymbol{\theta}_{t,s}) & \text{if } s = N - 1 \end{cases},
\end{aligned}
\tag{36}
$$

and

$$
\begin{aligned}
X_{t,s} &\triangleq w\left(\eta(1 - \eta L)\nabla F(\boldsymbol{\theta}_{t,s-1})^\top \boldsymbol{\varepsilon}_t(\boldsymbol{\theta}_{t,s-1}) + \frac{\eta^2 L}{2}\left\|\boldsymbol{\varepsilon}_t(\boldsymbol{\theta}_{t,s-1})\right\|^2\right) \\
&\quad - w^2\eta^2(1 - \eta L)^2 \left\|\nabla F(\boldsymbol{\theta}_{t,s-1})\right\|^2 \sigma^2(\boldsymbol{\theta}_{t,s-1}) .
\end{aligned}
\tag{37}
$$

According to our Lemma A.8 and the fact that each dimension of $\boldsymbol{\varepsilon}_t(\boldsymbol{\theta}_{t,s})$ follows an independent Gaussian distribution given Assump. 2, the following holds

$$
\begin{aligned}
\mathbb{E}\left[\exp\left(\sum_{t=1}^{T}\sum_{s=1}^{N} X_{t,s}\right)\right] &\leq \exp\left(\sum_{t=1}^{T}\sum_{s=1}^{N}\left(w^2\eta^2(1 - \eta L)^2\left\|\nabla F(\boldsymbol{\theta}_{t,s-1})\right\|^2 + \frac{w\eta^2 L}{2}\right)\sigma^2(\boldsymbol{\theta}_{t,s-1})\right. \\
&\quad\left. - \sum_{t=1}^{T}\sum_{s=1}^{N} w^2\eta^2(1 - \eta L)^2\left\|\nabla F(\boldsymbol{\theta}_{t,s-1})\right\|^2 \sigma^2(\boldsymbol{\theta}_{t,s-1}) + \frac{1}{4}\right) \\
&= \exp\left(\sum_{t=1}^{T}\sum_{s=1}^{N}\frac{w\eta^2 L}{2}\sigma^2(\boldsymbol{\theta}_{t,s-1}) + \frac{1}{4}\right) .
\end{aligned}
\tag{38}
$$

Based on Markov inequality, we have that

$$\mathbb{P}\left[\exp\left(\sum_{t=1}^{T}\sum_{s=1}^{N}\mathrm{X}_{t,s}\right) > \frac{1}{2\delta}\exp\left(\sum_{t=1}^{T}\sum_{s=1}^{N}\frac{w\eta^2 L}{2}\sigma^2(\boldsymbol{\theta}_{t,s-1})\right)\right]$$

$$\leq \frac{\mathbb{E}\left[\exp\left(\sum_{t=1}^{T}\sum_{s=1}^{N}\mathrm{X}_{t,s}\right)\right]}{\exp\left(\sum_{t=1}^{T}\sum_{s=1}^{N}w\eta^2 L\sigma^2(\boldsymbol{\theta}_{t,s-1})/2\right)/(2\delta)} \tag{39}$$

$$\leq \delta \,.$$

Therefore, with a probability of at least $1-\delta$,

$$\sum_{t=1}^{T}\sum_{s=1}^{N}\mathrm{X}_{t,s} \leq \sum_{t=1}^{T}\sum_{s=1}^{N}\frac{w\eta^2 L}{2}\sigma^2(\boldsymbol{\theta}_{t,s-1}) + \ln\left(\frac{1}{2\delta}\right)\,, \tag{40}$$

which leads to the following inequality with $w = w$

$$\sum_{t=1}^{T}\sum_{s=1}^{N}\left(\eta(1-\eta L)\nabla F(\boldsymbol{\theta}_{t,s-1})^{\top}\boldsymbol{\varepsilon}_t(\boldsymbol{\theta}_{t,s-1}) + \frac{\eta^2 L}{2}\left\|\boldsymbol{\varepsilon}_t(\boldsymbol{\theta}_{t,s-1})\right\|^2\right) \leq$$

$$\sum_{t=1}^{T}\sum_{s=1}^{N}\left(w\eta^2(1-\eta L)^2\left\|\nabla F(\boldsymbol{\theta}_{t,s-1})\right\|^2\sigma^2(\boldsymbol{\theta}_{t,s-1}) + \frac{\eta^2 L}{2}\sigma^2(\boldsymbol{\theta}_{t,s-1})\right) + \frac{1}{w}\ln\left(\frac{1}{2\delta}\right)\,. \tag{41}$$

Of note, for every proxy step based on SGD:

$$F(\boldsymbol{\theta}_{t,s})$$

$$\overset{(a)}{\leq} F(\boldsymbol{\theta}_{t,s-1}) + \nabla F(\boldsymbol{\theta}_{t,s-1})^{\top}(\boldsymbol{\theta}_{t,s} - \boldsymbol{\theta}_{t,s-1}) + \frac{L}{2}\left\|\boldsymbol{\theta}_{t,s} - \boldsymbol{\theta}_{t,s-1}\right\|^2$$

$$\overset{(b)}{=} F(\boldsymbol{\theta}_{t,s-1}) - \eta\nabla F(\boldsymbol{\theta}_{t,s-1})^{\top}(\nabla F(\boldsymbol{\theta}_{t,s-1}) - \boldsymbol{\varepsilon}_t(\boldsymbol{\theta}_{t,s-1})) + \frac{\eta^2 L}{2}\left\|\nabla F(\boldsymbol{\theta}_{t,s-1}) - \boldsymbol{\varepsilon}_t(\boldsymbol{\theta}_{t,s-1})\right\|^2$$

$$\overset{(c)}{=} F(\boldsymbol{\theta}_{t,s-1}) + \eta(1-\eta L)\nabla F(\boldsymbol{\theta}_{t,s-1})^{\top}\boldsymbol{\varepsilon}_t(\boldsymbol{\theta}_{t,s-1}) + \left(\frac{\eta^2 L}{2} - \eta\right)\left\|\nabla F(\boldsymbol{\theta}_{t,s-1})\right\|^2 + \frac{\eta^2 L}{2}\left\|\boldsymbol{\varepsilon}_t(\boldsymbol{\theta}_{t,s-1})\right\|^2$$

$$\tag{42}$$

where $(a)$ derives from the Lipschitz smoothness of function $F$ (i.e., Assump. 3), $(b)$ comes from the standard SGD update and the definition of $\boldsymbol{\varepsilon}_t(\boldsymbol{\theta}_{t,s})$, and $(d)$ is a rearrangement of the results in $(c)$.

By introducing the results above into (42) and choosing $w^{-1} = 2\beta\eta$ with $\beta \triangleq \max\{\kappa, \sigma^2\}$, we have

$$\sum_{t=1}^{T}\sum_{s=1}^{N}F(\boldsymbol{\theta}_{t,s}) \overset{(a)}{\leq} \sum_{t=1}^{T}\sum_{s=1}^{N}\left(F(\boldsymbol{\theta}_{t,s-1}) + \left(w\eta^2(1-\eta L)^2\sigma^2(\boldsymbol{\theta}_{t,s-1}) - \eta(1-\frac{\eta L}{2})\right)\left\|\nabla F(\boldsymbol{\theta}_{t,s-1})\right\|^2\right.$$

$$\left. + \frac{\eta^2 L}{2}\sigma^2(\boldsymbol{\theta}_{t,s-1})\right) + \frac{1}{w}\ln\left(\frac{1}{2\delta}\right)$$

$$\overset{(b)}{\leq} \sum_{t=1}^{T}\sum_{s=1}^{N}\left(F(\boldsymbol{\theta}_{t,s-1}) + \left(\frac{1}{2}\eta(1-\eta L)^2 - \eta(1-\frac{\eta L}{2})\right)\left\|\nabla F(\boldsymbol{\theta}_{t,s-1})\right\|^2\right.$$

$$\left. + \frac{\eta^2 L}{2}\sigma^2(\boldsymbol{\theta}_{t,s-1})\right) + 2\beta\eta\ln\left(\frac{1}{2\delta}\right)$$

$$\overset{(c)}{\leq} \sum_{t=1}^{T}\sum_{s=1}^{N}\left(F(\boldsymbol{\theta}_{t,s-1}) - \frac{\eta}{2}\left\|\nabla F(\boldsymbol{\theta}_{t,s-1})\right\|^2 + \frac{\eta^2 L}{2}\sigma^2(\boldsymbol{\theta}_{t,s-1})\right) + 2\beta\eta\ln\left(\frac{1}{2\delta}\right)$$

$$\tag{43}$$

where $(a)$ comes from $\eta \leq 1/L$, $(b)$ is due to the fact that $\sigma^2(\boldsymbol{\theta}_{t,s-1}) \leq \max\{\kappa, \sigma^2\} = \beta$, and $(c)$ is due to the fact that

$$
\begin{aligned}
\frac{\eta}{2}(1 - \eta L)^2 - \eta(1 - \frac{\eta L}{2}) &= \frac{\eta}{2}\left(1 - 2\eta L + \eta^2 L^2 - 2 + \eta L\right) \\
&= \frac{\eta}{2}\left(\eta^2 L^2 - \eta L - 1\right) \\
&\leq -\frac{\eta}{2} .
\end{aligned}
\tag{44}
$$

By rearranging the result in (43) and defining $\rho \triangleq (1 - \frac{1}{N})\frac{4\beta\gamma_{T_0}}{\sigma^2 T_0 d} + \frac{1}{N}$, we have

$$
\begin{aligned}
\frac{1}{NT}\sum_{t=1}^{T}\sum_{s=1}^{N}\left\|\nabla F(\boldsymbol{\theta}_{t,s-1})\right\|^2 &\leq \frac{2}{\eta NT}\left(F(\boldsymbol{\theta}_0) - F(\boldsymbol{\theta}_T)\right) + \frac{\eta L}{NT}\sum_{t=1}^{T}\sum_{s=1}^{N}\sigma^2(\boldsymbol{\theta}_{t,s-1}) + \frac{4\beta}{NT}\ln\left(\frac{1}{2\delta}\right) \\
&\leq \frac{2}{\eta NT}\left(F(\boldsymbol{\theta}_0) - \inf_{\boldsymbol{\theta}} F(\boldsymbol{\theta})\right) + \eta L\rho\sigma^2 + \frac{4\beta}{NT}\ln\left(\frac{1}{2\delta}\right)
\end{aligned}
\tag{45}
$$

where the last inequality comes from the fact that $F(\boldsymbol{\theta}_T) \leq \inf_{\boldsymbol{\theta}} F(\boldsymbol{\theta})$ and $\sigma^2(\boldsymbol{\theta}_{t,s-1}) \leq 4\beta\gamma_{T_0}/(T_0 d)$ in (33).

By choosing $T \geq \frac{2\Delta L}{N\sigma^2\rho}$ and $\eta = \sqrt{\frac{2\Delta}{NTL\sigma^2\rho}}$ where $\Delta \triangleq F(\boldsymbol{\theta}_0) - \inf_{\boldsymbol{\theta}} F(\boldsymbol{\theta})$, we conclude our proof.

**Remark 1.** The speedup achieved by OptEx matches that of basic sample averaging (i.e., data parallelism) for stochastic optimization with noisy gradients. However, the speedup from OptEx comes from reduced sequential iterations (first term on the RHS in (45)), while sample averaging derives from reduced gradient variance (second term on the RHS in (45)). When gradient noise is already small or in deterministic optimization (e.g., the experiments in Sec. 6.1), data parallelism may not provide noticeable speedup, but OptEx can still contribute significantly. Overall, OptEx works in a complementary direction to existing parallelization methods, including sample averaging, to speed up first-order optimization, especially when other methods are not applicable or underperforming as discussed in our Sec. 2).

## A.4 Proof of Theorem 3

We follow the idea in [37] to prove our Thm. 3. We first introduce the following lemma:

**Lemma A.9** (Lemma B.12 in [44]). *Let* $X_i \sim \mathcal{N}(0,1)$ *independently,* $Z \triangleq \sum_{i=1}^{n} X_i^2$, *and* $\epsilon \in (0,1)$ *then*

$$
\mathbb{P}\left(Z \leq (1 - \epsilon)n\right) \leq \exp\left(-\frac{n\epsilon^2}{6}\right) .
\tag{46}
$$

*Proof of Thm. 3.* When $\eta \in \left[1/(\sqrt{NT}L), 1/L\right]$, We consider the function

$$
F(\boldsymbol{\theta}) = \frac{L}{2}\left\|\boldsymbol{\theta}\right\|^2
\tag{47}
$$

where $\boldsymbol{\theta}_0$ is initialized with $\mathcal{N}(\mathbf{0}, \frac{\Delta}{L}\mathbf{I})$.

We abuse $\varepsilon(\boldsymbol{\theta}_\tau)$ to denote the $\varepsilon(\boldsymbol{\theta}_{t,s-1})$ defined in our (36). Based on the update rule of stochastic gradient descent, we then have that

$$
\begin{aligned}
\boldsymbol{\theta}_\tau &= \boldsymbol{\theta}_{\tau-1} - \eta\left(L\boldsymbol{\theta}_{\tau-1} + \varepsilon(\boldsymbol{\theta}_{\tau-1})\right) \\
&= (1 - \eta L)\boldsymbol{\theta}_{\tau-1} - \eta\varepsilon(\boldsymbol{\theta}_{\tau-1}) \\
&= (1 - \eta L)^\tau \boldsymbol{\theta}_0 + \sum_{i=0}^{\tau-1}\eta(1 - \eta L)^{\tau-i-1}\varepsilon(\boldsymbol{\theta}_i) .
\end{aligned}
\tag{48}
$$

Since $\varepsilon(\boldsymbol{\theta}_i)$ follows $\mathcal{N}(\mathbf{0}, \sigma^2(\boldsymbol{\theta}_i))$ independently where we abuse $\sigma^2(\boldsymbol{\theta}_i)$ to denote the $\sigma^2(\boldsymbol{\theta}_{t,s-1})$ defined in (36) and $\boldsymbol{\theta}_0$ is initialized with $\mathcal{N}(\mathbf{0}, \frac{\Delta}{L}\mathbf{I})$, we then have that

$$\boldsymbol{\theta}_\tau \sim \mathcal{N}\left(\mathbf{0}, \left((1-\eta L)^{2\tau}\frac{\Delta}{L} + \sum_{i=0}^{\tau-1}\eta^2(1-\eta L)^{2(\tau-i-1)}\sigma^2(\boldsymbol{\theta}_i)\right)\mathbf{I}\right). \tag{49}$$

Let $\delta \in (0,1)$ and $\widetilde{\beta} \triangleq \min\{1/(1+1/\sigma^2)^{T_0}\kappa, \sigma^2\}$, since $\|\nabla F(\boldsymbol{\theta}_\tau)\|^2 = L^2\|\boldsymbol{\theta}_\tau\|^2$, by introducing the results above into Lemma A.9 with $\epsilon = 1/2$ and $d \geq d_0 \triangleq 24\ln(NT/\delta)$, with a probability of at least $1-\delta$,

$$\begin{aligned}
\min_{\tau\in[NT]}\|\nabla F(\boldsymbol{\theta}_\tau)\|^2 &= \frac{1}{NT}\sum_{\tau=1}^{NT}\|\nabla F(\boldsymbol{\theta}_\tau)\|^2 \\
&\geq \frac{dL^2}{2}\left((1-\eta L)^{2\tau}\frac{\Delta}{L} + \sum_{i=0}^{\tau-1}\eta^2(1-\eta L)^{2(\tau-i-1)}\sigma^2(\boldsymbol{\theta}_i)\right) \\
&\geq \frac{dL^2}{2}\left((1-\eta L)^{2\tau}\frac{\Delta}{L} + \sum_{i=0}^{\tau-1}\eta^2(1-\eta L)^{2(\tau-i-1)}\widetilde{\beta}\right) \\
&= \frac{dL^2}{2}\left((1-\eta L)^{2\tau}\frac{\Delta}{L} + \frac{1-(1-\eta L)^{2\tau}}{1-(1-\eta L)^2}\eta^2\widetilde{\beta}\right) \\
&= \frac{d}{2}\left((1-\eta L)^{2\tau}\Delta L + \left(1-(1-\eta L)^{2\tau}\right)\frac{\eta L\widetilde{\beta}}{2-\eta L}\right) \\
&\geq \frac{d}{2}\min\left\{\Delta L, \frac{\eta L\widetilde{\beta}}{2-\eta L}\right\} \\
&\geq \frac{d}{2}\min\left\{\Delta L, \frac{\widetilde{\beta}}{2\sqrt{NT}}\right\} \\
&\geq \frac{d_0\min\left\{\Delta L, \widetilde{\beta}\right\}}{4\sqrt{NT}}.
\end{aligned} \tag{50}$$

When $\eta \in \left[0, 1/(\sqrt{NT}L)\right]$, we consider the function

$$F(\boldsymbol{\theta}) = \frac{1}{4\max\left\{1/L, \sum_{\tau=1}^{NT}\eta\right\}}\left\|\boldsymbol{\theta}^\top\boldsymbol{e}_1\right\|^2 \tag{51}$$

where $\boldsymbol{\theta}_0$ is initialized with $\boldsymbol{\theta}_0^\top = \left[\sqrt{d\Delta\max\left\{1/L, \sum_{\tau=1}^{NT}\eta\right\}}, 0, \cdots, 0\right]$.

Similarly, we have

$$\boldsymbol{\theta}_\tau = \left(1 - \frac{1}{2\max\left\{1/L, \sum_{\tau=1}^{NT}\eta\right\}}\right)^\tau\boldsymbol{\theta}_0 + \sum_{i=0}^{\tau-1}\eta\left(1 - \frac{1}{2\max\left\{1/L, \sum_{\tau=1}^{NT}\eta\right\}}\right)^{\tau-i-1}\varepsilon(\boldsymbol{\theta}_i), \tag{52}$$

and

$$\boldsymbol{\theta}_\tau \sim \mathcal{N}\left(\left(1 - \frac{1}{2\max\left\{1/L, \sum_{\tau=1}^{NT}\eta\right\}}\right)^\tau\boldsymbol{\theta}_0, \left(\sum_{i=0}^{\tau-1}\eta^2\left(1 - \frac{1}{2\max\left\{1/L, \sum_{\tau=1}^{NT}\eta\right\}}\right)^{2(\tau-i-1)}\sigma^2(\boldsymbol{\theta}_i)\right)\mathbf{I}\right). \tag{53}$$

Therefore, let $\boldsymbol{\theta}_\tau^{(1)}$ denote the first element of $\boldsymbol{\theta}_\tau$, we have

$$
\mathbb{E}\left[\boldsymbol{\theta}_\tau^{(1)}\right] \overset{(a)}{=} \left(1 - \frac{1}{2\max\left\{1/L, \sum_{\tau=1}^{NT}\eta\right\}}\right)^\tau \sqrt{d\Delta\max\left\{1/L, \sum_{\tau=1}^{NT}\eta\right\}}
$$

$$
\overset{(b)}{\geq} \sqrt{d\Delta\max\left\{1/L, \sum_{\tau=1}^{NT}\eta\right\}}\exp\left(\left(\ln\frac{1}{2}\right)\sum_{\tau=1}^{NT}\frac{\eta}{\max\left\{1/L, \sum_{\tau=1}^{NT}\eta\right\}}\right) \qquad (54)
$$

$$
\overset{(c)}{\geq} \frac{1}{2}\sqrt{d\Delta\max\left\{1/L, \sum_{\tau=1}^{NT}\eta\right\}}
$$

where $(b)$ comes from the fact that $1 - z/2 \geq \exp(\ln(1/2)z)$ for all $z \in [0,1]$.

In addition, we have

$$
\mathrm{var}\left[\boldsymbol{\theta}_\tau^{(1)}\right] = \sum_{i=0}^{\tau-1}\eta^2\left(1 - \frac{1}{2\max\left\{1/L, \sum_{\tau=1}^{NT}\eta\right\}}\right)^{2(\tau-i-1)}\sigma^2(\boldsymbol{\theta}_i^{(1)})
$$

$$
\leq \sum_{i=0}^{\tau-1}\eta^2\beta \qquad (55)
$$

$$
= \frac{\beta}{L^2}
$$

where $\beta \triangleq \max\{\kappa, \sigma^2\}$.

Let $\Psi$ denote the CDF of standard normal distribution and follow the idea in [37], by choosing

$$
d > d_0 \triangleq \frac{16\beta/L^2\left(\Psi^{-1}\left(1 - \frac{\delta}{NT}\right)\right)^2}{\Delta\max\left\{1/L, \sum_{\tau=1}^{NT}\eta\right\}} = \mathcal{O}\left(\beta/(\Delta L^2)\ln NT/\delta\right), \qquad (56)
$$

then

$$
\mathbb{P}\left(\frac{\boldsymbol{\theta}_\tau^{(1)} - \mathbb{E}\left[\boldsymbol{\theta}_\tau^{(1)}\right]}{\sqrt{\mathrm{var}\left[\boldsymbol{\theta}_\tau^{(1)}\right]}} \geq -\frac{\frac{1}{4}\sqrt{d\Delta\max\left\{1/L, \sum_{\tau=1}^{NT}\eta\right\}}}{\sqrt{\beta/L^2}}\right)
$$

$$
\geq \mathbb{P}\left(\frac{\boldsymbol{\theta}_\tau^{(1)} - \mathbb{E}\left[\boldsymbol{\theta}_\tau^{(1)}\right]}{\sqrt{\mathrm{var}\left[\boldsymbol{\theta}_\tau^{(1)}\right]}} \geq -\frac{\frac{1}{4}\sqrt{d\Delta\max\left\{1/L, \sum_{\tau=1}^{NT}\eta\right\}}}{\sqrt{\beta/L^2}}\right) \qquad (57)
$$

$$
= 1 - NT\delta.
$$

That is, with a probability of at least $1 - \delta/(NT)$,

$$
\boldsymbol{\theta}_\tau^{(1)} \geq \frac{1}{4}\sqrt{d\Delta\max\left\{1/L, \sum_{\tau=1}^{NT}\eta\right\}}. \qquad (58)
$$

Since $\|\nabla F(\boldsymbol{\theta}_\tau)\|^2 = \left( \frac{1}{2\max\{1/L, \sum_{\tau=1}^{NT} \eta\}} \right)^2 \left\| \boldsymbol{\theta}^\top \boldsymbol{e}_1 \right\|^2$, we conclude our proof by applying union bound on (58) as below

$$\min_{\tau \in [NT]} \|\nabla F(\boldsymbol{\theta}_\tau)\|^2 = \frac{1}{NT} \sum_{\tau=1}^{NT} \|\nabla F(\boldsymbol{\theta}_\tau)\|^2$$
$$\geq \frac{d_0 \Delta}{4\max\left\{1/L, \sum_{\tau=1}^{NT} \eta\right\}} \tag{59}$$
$$\geq \frac{d_0 \Delta L}{4\sqrt{NT}}$$

where the last inequality comes from the fact that $\eta \in [0, 1/(\sqrt{NT}L)]$. This finally concludes our proof.

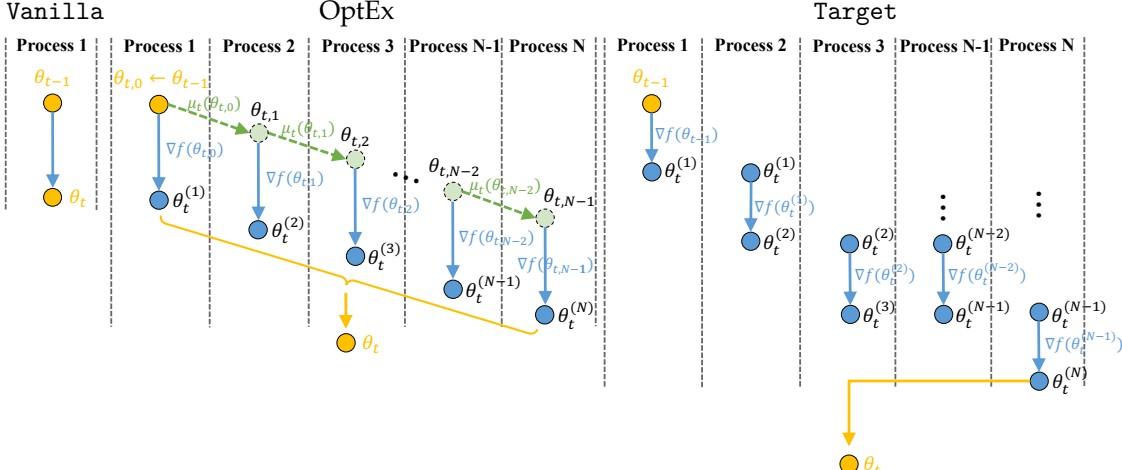

Figure 5: An illustrated comparison among our OptEx and all the baselines at iteration $t$.

## Appendix B    Experiments

### B.1    Baselines

In this section, we provide an illustrated comparison between our OptEx and all the baselines at iteration $t$ in Fig. 5. Notably, the `Target` baseline represents an ideal parallelization of the `Vanilla` baseline. However, this is impractical because the ground-truth gradient (i.e., $\nabla f(\cdot)$) required by a process $i \in [N]$ to produce the update can not be obtained before the start of this process. More specifically, this gradient is the outcome at the end of the corresponding process. In contrast, our OptEx framework makes use of the kernelized gradient estimation (i.e., $\boldsymbol{\mu}_t(\cdot)$) to achieve approximated parallelized iterations for FOO as illustrated in our Fig. 5, which is more practical and useful.

### B.2    Settings

#### B.2.1    Optimization of Synthetic Functions

Let input $\boldsymbol{\theta} = [\theta_i]_{i=1}^d$, the Ackley, Sphere, and Rosenbrock functions applied in our synthetic experiments are given below, which have been slightly modified compared with the standard ones.

$$F(\boldsymbol{\theta}) = -20 \exp\left(-0.2\sqrt{\frac{1}{d}\sum_{i=1}^d \theta_i^2}\right) - \exp(\frac{1}{d}\sum_{i=1}^d \cos\left(2\pi\theta_i\right)) + 20 + \exp(1), \text{(Ackley)}$$

$$F(\boldsymbol{\theta}) = \sqrt{\frac{1}{d}\sum_{i=1}^d \theta_i^2}, \text{(Sphere)} \tag{60}$$

$$F(\boldsymbol{\theta}) = \frac{1}{d}\sum_{i=1}^{d-1}\left[100(\theta_{i+1} - \theta_i)^2 + (1 - \theta_i)^2\right], \text{(Rosenbrock)}$$

Note that both Ackley and Sphere function achieve their minimum (i.e., $\min F(\boldsymbol{\theta}) = 0$) at $\boldsymbol{\theta}^* = \mathbf{0}$, whereas Rosenbrock function achieves its minimum (i.e., $\min F(\boldsymbol{\theta}) = 0$) at $\boldsymbol{\theta}^* = \mathbf{1}$.

In this experiment, the parallelism of $N = 5$ is applied and all the baselines introduced in Sec. 6.1 as well as our OptEx are based on Adam [4] with a learning rate of 0.1, $\beta_1 = 0.9$, and $\beta_2 = 0.999$. In addition, we employ a Matérn kernel-based gradient estimation in our OptEx with $T_0 = 20$.

#### B.2.2    Optimization of Reinforcement Learning Tasks

Our experimental framework is built on the Deep Q-Network (DQN) algorithm, as outlined in [39], and implemented within the OpenAI Gym environment [38]. This study investigates the effectiveness of different optimizer configurations across classical discrete control tasks provided by Gym.

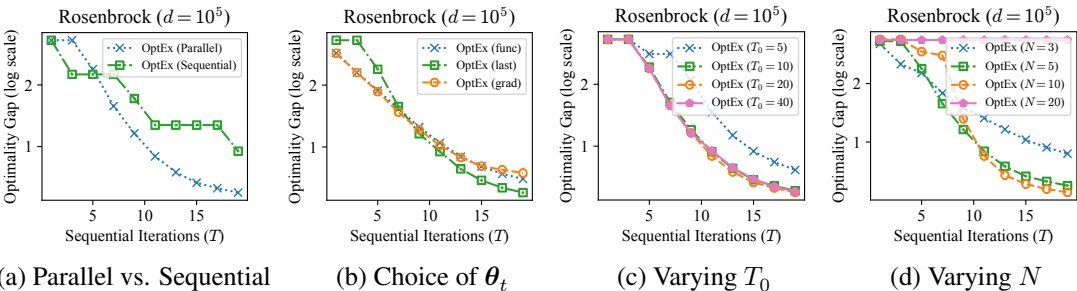

Figure 6: Ablation studies on the Rosenbrock synthetic function.

Each trial is conducted on a dedicated CPU to maintain consistency in computational conditions. The DQN architecture consists of dual fully connected layers, with 64 or 128 neurons tailored to each task's requirements. Hyperparameters, including a learning rate of 0.001, a reward discount factor of 0.95, and a batch size of 256, are applied for fairness and consistency across experiments.

Performance evaluation of the optimizer-enhanced DQN agents is systematically carried out over 100 to 200 episodes per game, employing an $\epsilon$-greedy policy with a minimum epsilon of 0.1 and an exponential epsilon decay with a rate of $2^{-\frac{1}{1500}}$. A preliminary warm-up phase of either 30 or 50 episodes, depending on the task, is incorporated to stabilize initial learning dynamics. Besides, all baselines introduced in Sec.6.1 and our OptEx are based on Adam[4] with a learning rate of 0.001, $\beta_1 = 0.9$, and $\beta_2 = 0.999$. For OptEx, we utilize a Mat'ern kernel-based gradient estimation, with $T_0 = 150$ to accommodate the variance in RL tasks, and parallelism of $N = 4$ is applied.

### B.2.3  Optimization of Neural Network Training

In this experiment, we compared our OptEx with other baselines using both image classification and text autoregression tasks. Here, we simply make use of the `jax.vmap` function to simulate parallel computing and measure the wallclock time for each sequential iteration. We believe that the time efficiency of our OptEx can be further improved when it is more properly implemented on a parallel computing platform. Besides, to reduce the computational cost of our kernelized gradient estimation in these high-dimensional optimization problems, we propose to use a randomly sampled subset of dimensions (e.g., $\widetilde{d} = 10^4$ for image classification and $\widetilde{d} = 10^5$ for text autoregression) from the total $d$ dimension to compute the kernel value $k(\cdot, \cdot)$ in each sequential iteration of our OptEx.

**Image Classification.**  In this image classification task, we train a 9-layer MLP (including input and output layer) with skip connections on MNIST [45], Fashion MNIST [46] and a 10-layer MLP (including input and output layer) with skip connections on CIFAR-10 [41] datasets, which have a parameter size of $d = 978186$ for (fashion-)MNIST and $d = 2412298$ for CIFAR-10. Both our OptEx and other baselines are based on SGD [1] with a learning rate of 0.001, a batch size of 512, and parallelism of $N = 4$. For OptEx, we employ a Matérn kernel-based gradient estimation with $T_0 = 6$.

**Text Autoregression.**  In addition, we further train a simple transformer from Haiku library [42] with a parameter size of $d = 1626496$ on the corpus of "Harry Potter and the Sorcerers Stone" and a subset work from Shakespeare. In both tasks, all the baselines introduced in Sec. 6.1 and our OptEx are based on SGD [1] with a learning rate of 0.01, batch size of 256 and parallelism of $N = 4$. For OptEx, we employ a Matérn kernel-based gradient estimation, where $T_0 = 10$.

### B.3  More Results

**Ablation Studies on Synthetic Function.**  To better understand our OptEx algorithm, we have conducted a number of ablation studies on the Rosenbrock synthetic function with a dimension of $d = 10^5$. The results are in Fig. 6, in which there are 4 different types of comparisons: (a) We have compared our OptEx with vs. without evaluating the intermediate gradients, i.e., $\left\{\nabla f(\boldsymbol{\theta}_{t,i-1})\right\}_{i=1}^{N-1}$ at every iteration $t$, denoting as `parallel` and `sequential` respectively in Fig. 6 (a), which aims to

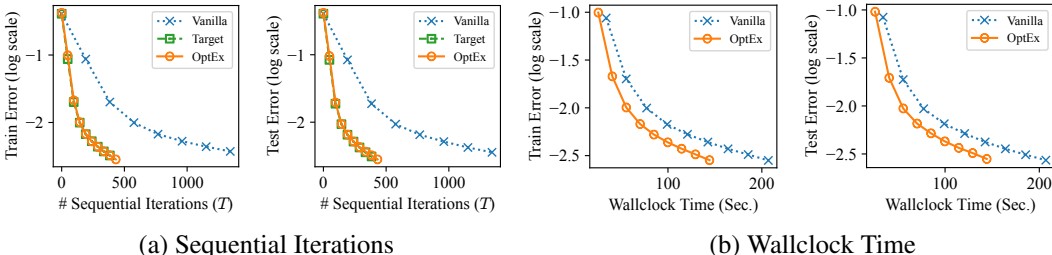

(a) Sequential Iterations           (b) Wallclock Time

Figure 7: Comparison of the train and test error (i.e., 1 - accuracy in log scale for $y$-axis) achieved by different optimizers when training MLP with residual connections on MNIST dataset with (a) a varying number $T$ of sequential iteration and (b) a varying wallclock time ($x$-axis). The parallelism $N$ is set to 4 and each curve denotes the mean from 5 independent runs. The wallclock time is evaluated on an AMD EPYC 7763 CPU.

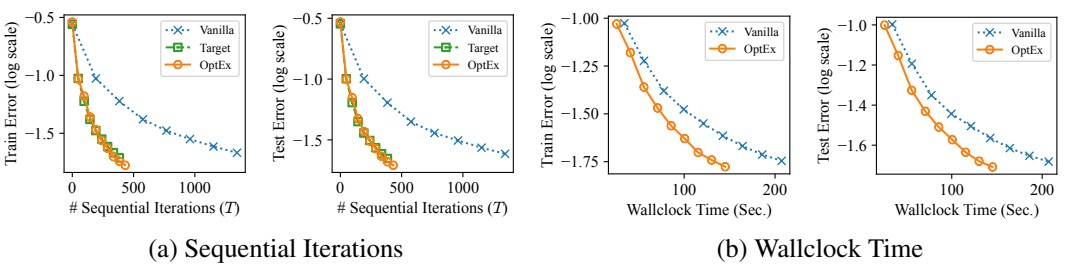

(a) Sequential Iterations           (b) Wallclock Time

Figure 8: Comparison of the train and test error (i.e., 1 - accuracy in log scale for $y$-axis) achieved by different optimizers when training MLP with residual connections on the fashion-MNIST dataset with (a) a varying number $T$ of sequential iteration and (b) a varying wallclock time ($x$-axis). The parallelism $N$ is set to 4 and each curve denotes the mean from 5 independent runs. Similarly, the wallclock time is evaluated on an AMD EPYC 7763 CPU.

show the importance of these intermediate gradients on an accurate gradient estimation and therefore improved convergence of our OptEx as justified in our Sec. 4.3. (b) We have compared our OptEx using different principles to choose $\boldsymbol{\theta}_t$ from $\{\boldsymbol{\theta}_t^{(i)}\}_{i=1}^N$, including using function value (denoted as func in Fig. 6 (b)) via $\boldsymbol{\theta}_t = \arg\min_{\boldsymbol{\theta} \in \{\boldsymbol{\theta}_t^{(i)}\}_{i=1}^N} f(\boldsymbol{\theta})$, using $\boldsymbol{\theta}$ from the process $N$ (denoted as last in Fig. 6 (b), i.e., the standard principle in Algo. 1) with $\boldsymbol{\theta}_t = \boldsymbol{\theta}_t^{(N)}$, and using gradient norm (denoted as grad in Fig. 6 (b)) via $\boldsymbol{\theta}_t = \arg\min_{\boldsymbol{\theta} \in \{\boldsymbol{\theta}_t^{(i)}\}_{i=1}^N} \|\nabla f(\boldsymbol{\theta})\|$. (c) We have compared our OptEx with varying $T_0$ in Fig. 6 (c). (d) We have compared our OptEx with varying $N$ in Fig. 6 (d). All the other experimental settings follow from the same ones in our Appx. B.2.1.

The results presented in Fig. 6 (a) indicate that evaluating intermediate gradients $\left\{\nabla f(\boldsymbol{\theta}_{t,i-1})\right\}_{i=1}^{N-1}$ at each iteration $t$ is crucial for achieving better convergence with our OptEx. This improved performance likely stems from these evaluations being more aligned with the gradient approximations required at point $\boldsymbol{\theta}$ in our OptEx, which is essential to achieve accurate gradient estimation and therefore well-performing convergence in our OptEx. Consequently, these findings underscore the importance and necessity of line 7 in Algo. 1, as discussed in Sec. 4.3. Further, Fig. 6 (b) shows that utilizing $\boldsymbol{\theta}$ from the final process $N$ (denoted as last) where $\boldsymbol{\theta}_t = \boldsymbol{\theta}_t^{(N)}$, typically results in marginally better convergence. This approach maximizes the benefits of parallelism within $N$ processes, unlike the other methods which often operate under reduced parallelism due to constraints in optimizing $\boldsymbol{\theta}_t^{(N)}$. Additionally, Fig. 6 (c) reveals that maintaining a gradient history length of $T_0 \leq 10$ generally improves convergence. Extending $T_0$ beyond 10, however, does not significantly improve outcomes, which thereby validates our theoretical insights from Sec. 5.1. Finally, Fig. 6 (d) shows that increasing the number of processes when $N \leq 10$ improves the iteration complexity of our OptEx. However, as $N$ increases to 20, convergence deteriorates. This observation aligns with the theoretical insights in our Sec. 5.2, which posits that while increasing $N$ up to an optimal point $N_{\text{opt}} = \Delta\eta^2/(LT\sigma^2\rho)$ enhances convergence, further increases can degrade performance.

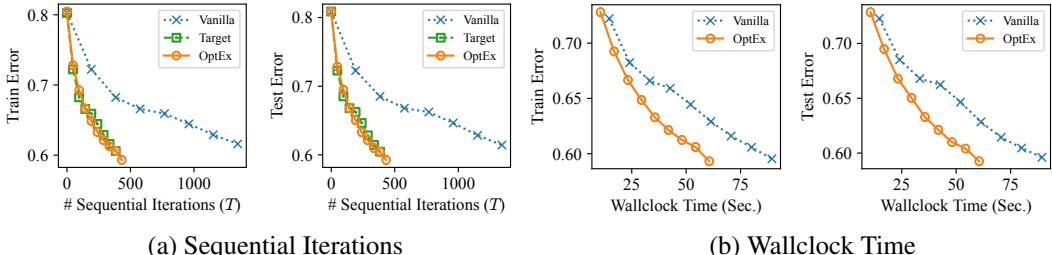

(a) Sequential Iterations        (b) Wallclock Time

Figure 9: Comparison of the train and test error (i.e., 1 - accuracy in log scale for $y$-axis) achieved by different optimizers when training MLP with residual connections on CIFAR-10 dataset with (a) a varying number $T$ of sequential iteration and (b) a varying wallclock time ($x$-axis). The parallelism $N$ is set to 4 and each curve denotes the mean from 5 independent runs. Similarly, the wallclock time is evaluated on a single NVIDIA RTX 4090 GPU.

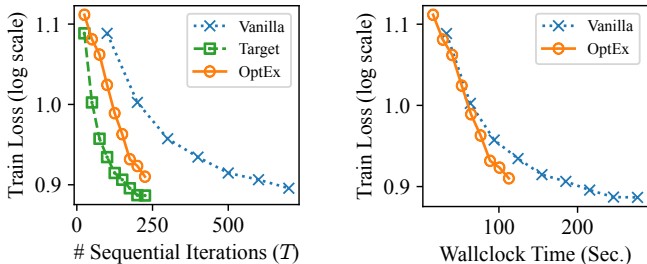

Figure 10: Comparison of the training loss ($y$-axis) achieved by different optimizers when training transformer on the corpus of "Harry Potter and the Sorcerer's Stone" with a varying number $T$ of sequential iteration and a varying wallclock time ($x$-axis). The parallelism $N$ is set to 4 and each curve denotes the mean from 3 independent experiments. The wallclock time is evaluated on a single NVIDIA RTX 4090 GPU.

**Image Classification on MNIST and Fashion-MNIST.** We have also compared the training and test errors achieved by different optimizers when training an MLP with residual connections on the MNIST and Fashion-MNIST datasets. The results in Fig.7 and Fig.8 indicate that our OptEx consistently improves over the Vanilla baseline and performs comparably to the Target baseline in terms of the number of sequential iteration required to achieve the same level of training or test error. Furthermore, our OptEx significantly improves the time efficiency of training the MLP on both datasets, as evidenced by the results in Fig.7 and Fig.8. These findings hence also validate the efficacy of our OptEx in accelerating FOO across various image classification tasks.

**Image Classification on CIFAR-10.** Besides the test errors presented in Sec. 6.3, we also provide a comprehensive comparison of both training and test errors achieved by different optimizers when training an MLP with residual connections on the CIFAR-10 dataset. This comparison, shown in Fig.9, considers varying numbers $T$ of sequential iterations and varying wallclock time. Due to the computational cost of training deep neural networks on CIFAR-10, we evaluate the wallclock time for all optimizers using a single NVIDIA RTX 4090 GPU. Notably, similar to the results in Sec. 6.3, our OptEx consistently outperforms the Vanilla baseline and performs comparably to the Target baseline in both training and test errors. Additionally, our OptEx considerably enhances the time efficiency of training the MLP on CIFAR-10, as demonstrated by the results in Fig.9. These results therefore adequately verify the efficacy of our OptEx in expediting FOO.

**Autoregression on Text Corpus.** In addition to the training results on the Shakespeare corpus, we also present the training loss achieved by different optimizers when training the same transformer on the corpus of "Harry Potter and the Sorcerer's Stone" in Fig.10. Remarkably, Fig.10 shows that our OptEx still consistently outperforms the Vanilla baseline and performs comparably to the Target baseline. Moreover, the results in Fig. 10 demonstrate that our OptEx significantly enhances the time efficiency of training the transformer on the Harry Potter corpus. These findings thus further validate the efficacy of our OptEx in accelerating FOO across various types of learning tasks.

