# OpenReview forum: "OptEx: Expediting First-Order Optimization with Approximately Parallelized Iterations"
_NeurIPS.cc/2024/Conference — NeurIPS 2024 poster_

### Official Review · Reviewer_YWBT · 2024-07-10

**Soundness:** 4
**Presentation:** 4
**Contribution:** 3
**Rating:** 7
**Confidence:** 3

**Summary:**

The paper proposes a general framework for accelerating first-order optimization methods. The framework leverages parallel computing by estimating the gradients using kernel methods and breaking iterative dependencies. The paper establishes theoretical guarantees that the method gives an acceleration rate of $\sqrt{N}$ for $N$ parallelism. Then the paper does extensive empirical studies to show the improvements of this framework.

**Strengths:**

1. Very clearly written. It's immediately clear from the abstract and introduction what kind of work the authors are doing and in the main body the details are given in a very pleasant manner with beautiful plots and formula. The whole paper is well organized with theory and practice covered in style.
2. Novelty. I'm quickly convinced that the paper's contribution is novel. The related work session well summarizes previous work and it's clear to me that the paper investigates a direction that is distinctively different and important.
3. A complete story. The paper studies both theoretically and empirically the aspect, with both results agreeing with each other to a certain extent, forming a complete story.

**Weaknesses:**

The theoretical guarantees rely on assumptions that might not work for $N$ very large, thus limit the extent to which the framework can accelerate optimization.

In the empirical part, $N$ are not very large, thus unable to show that the framework can accelerate things to great extent in general.

**Questions:**

Suppose that the step size is very small, can the $N$ potentially be very large?

**Limitations:**

The authors adequately addressed the limitations.

---

> ### Author Rebuttal · Authors · 2024-08-07
>
> We are grateful to Reviewer YWBT for the positive and constructive feedback! We appreciate that the reviewer highly recognizes that our paper is **very well written**, its  contribution is **novel**, and our paper forms a **complete story with both theoretical and empirical supports agreeing with each other**. We would like to address your comments as follows.
>
> ---
>
> > The theoretical guarantees rely on assumptions that might not work for $N$ very large, thus limit the extent to which the framework can accelerate optimization.
> > In the empirical part, $N$ are not very large, thus unable to show that the framework can accelerate things to great extent in general.
> > Suppose that the step size is very small, can the $N$ potentially be very large?
>
> Thank you for your insightful comments. We acknowledge that when a relatively large learning rate $\eta$ is applied, the parallelism $N$ of our OptEx framework should not be excessively large to ensure fast convergence to a stationary point, as indicated in line 550 and Equation (45). In this scenario, the error of our kernelized gradient estimation, which is related to $\rho$ that requires a smaller $N$ to achieve a smaller value, will dominate the convergence of OptEx (refer to the second term on the RHS in Equation 45).
>
> However, when the learning rate (i.e., the step size) is very small, $N$ can indeed be potentially very large. In this different scenario, $\frac{2\Delta}{\eta NT}$ (refer to the first term on the RHS in Equation 45) will dominate the convergence of OptEx. Consequently, we can choose a very large $N$ to enjoy a small value of $\frac{2\Delta}{\eta NT}$ and thus achieve fast convergence of OptEx according to Equation 45.
>
> We support this with additional experimental results following the same setup as in Section 6.1. Using Adam with $\eta=0.001$,  $T_0=50$, and $N=100$ on Ackley function ($d=10^5$), we observe that when $\eta$ is small (0.001 vs. 0.1), a large parallelism $N$ (100 vs. 5) can be applied, and our OptEx remains enjoying a speedup of approximately $\sqrt{N}$.
>
> |         | $T=50$ | $T=100$ | $T=150$ | $T=200$ |
> |---------|--------|---------|---------|---------|
> | Vanilla | 3.0844 | 2.1665  | 1.2413  | **0.5220**  |
> |         | $T=5$  | $T=10$  | $T=15$  | $T=20$  |
> | OptEx   | 0.7578 | 0.7016  | **0.4696**  | 0.2383  |
>
> ---
>
> Thanks for your insightful questions. We will incorporate these discussions above into our revised version. We hope our clarification will address your concerns and improve your opinion of our work.

---

> > ### Comment · Reviewer_YWBT · 2024-08-07
> > **Very Good Rebuttal**
> >
> > Thanks the authors for their well-written rebuttal. The response on my confusion about N is very clear. Thanks a lot.

---

> ### Author Response · Authors · 2024-08-08
>
> Dear Reviewer YWBT,
>
> Thank you so much for your prompt and positive response after our rebuttal! We are so happy to hear that your concerns have been well addressed. We will incorporate the discussions above into our revised version.
>
> Sincerely, Authors

---

### Official Review · Reviewer_1m6n · 2024-07-10

**Soundness:** 3
**Presentation:** 3
**Contribution:** 3
**Rating:** 6
**Confidence:** 3

**Summary:**

The paper "OptEx: Expediting First-Order Optimization with Approximately Parallelized Iterations" introduces a novel framework, OptEx, designed to improve the efficiency of first-order optimization (FOO) algorithms by approximately parallelizing their iterations.

**Strengths:**

1. The introduction of a general framework for parallelizing iterations in FOO is novel and addresses a significant inefficiency in traditional optimization methods.
2. The paper provides robust theoretical guarantees for the performance of the OptEx framework, including bounds on estimation error and iteration complexity.
3. The extensive experiments across various domains (synthetic functions, reinforcement learning, and neural network training) demonstrate the practical applicability and efficiency gains of OptEx.

**Weaknesses:**

1. The kernelized gradient estimation and the associated computational techniques are complex, which might pose challenges for practical implementation and understanding by a broader audience.
2. The efficiency and scalability of OptEx in very large-scale optimization problems or in highly dynamic environments might be limited
3. The theoretical guarantees rely on certain assumptions (e.g., the Gaussian distribution of gradients), which may not hold in all practical scenarios, potentially limiting the generality of the results.

**Questions:**

see weakness

---

> ### Author Rebuttal · Authors · 2024-08-07
>
> We thank Reviewer 1m6n for recognizing that our OptEx framework is **novel**, it addresses a **significant inefficiency** in traditional optimization methods, and it has **robust theoretical guarantees** and **extensive empirical results** to support its **practical applicability** and **efficiency gains**. We would like to address your concerns below.
>
> ---
>
> > The kernelized gradient estimation and the associated computational techniques are complex, which might pose challenges for practical implementation and understanding by a broader audience.
>
> Thank you for your feedback. We would like to emphasize that our kernelized gradient estimation and the associated computational techniques are, in fact, quite simple and easy to understand. They involve computations similar to those found in kernel regression (as detailed in Proposition 4.1). Furthermore, our implementation of these techniques is concise, requiring only 16 lines of code (excluding blank lines), which we have provided in our supplementary materials. By following our implementations, we believe that the audience should be able to deploy our method to their problems without significant challenges.
>
> > The efficiency and scalability of OptEx in very large-scale optimization problems or in highly dynamic environments might be limited.
>
> Thank you for raising this point. The efficiency and scalability of OptEx in **relatively large-scale** optimization problems have been well demonstrated by our results on neural networks with millions of parameters in Section 6.3. These results adequately highlight the potential of the OptEx framework in accelerating first-order optimization by approximately parallelizing its iterations.
>
> Regarding the **very large-scale** optimization problems and **highly dynamic environments** you mentioned, we would appreciate it if you could provide specific examples for us to explore in future research since different practical scenarios often require specialized adaptations of our OptEx framework to achieve a compelling performance. Overall, we find these directions quite interesting and worthwhile to investigate further. Thank you for bringing them to our attention!
>
> > The theoretical guarantees rely on certain assumptions (e.g., the Gaussian distribution of gradients), which may not hold in all practical scenarios, potentially limiting the generality of the results.
>
> Thank you for your insightful comment. If you are referring to our Assumption 1, we would like to clarify that our assumption pertains to the **gradient noise** rather than the **gradient itself** following a Gaussian distribution when dealing with the stochastic optimization problem in Equation 1. This assumption has been widely recognized and applied in the literature [18, 19, 20].
>
> However, if you are referring to our Assumption 2,  we agree with you that Assumption 2 (i.e., $\nabla F$ is sampled from a Gaussian prior $GP(0, k(\cdot,\cdot))$) may not hold in all practical scenarios given a predefined kernel $k$, as $k$ may not be the ground truth. Nonetheless, our extensive experiments on synthetic functions, reinforcement learning tasks, and neural network training, as presented in Section 6, have demonstrated that our OptEx framework is in fact quite robust with a predefined kernel $k$ in practice. These experiments show that OptEx can achieve reasonably good performance across various real-world problems, even when Assumption 2 may not necessarily hold.
>
> We believe these empirical results support the generality of OptEx. If you have any further concerns regarding the generality, we would be happy to address them.
>
> ---
>
> We thank the reviewer for the valuable input and hope our answers can increase your opinions of our work.

---

> ### Comment · Reviewer_1m6n · 2024-08-10
> **response to author**
>
> Thanks for the detailed response from the author, my concern has already been addressed.

---

> > ### Author Response · Authors · 2024-08-10
> >
> > Dear Reviewer 1m6n,
> >
> > Thank you so much for your positive response after our rebuttal! We are so happy to hear that your concerns have been well addressed. We will incorporate all the discussions above into our revised version.
> >
> > Sincerely, Authors

---

### Official Review · Reviewer_ieTR · 2024-07-18

**Soundness:** 2
**Presentation:** 2
**Contribution:** 2
**Rating:** 3
**Confidence:** 4

**Summary:**

This paper presents OptEx, an approach to parallelize optimization methods by using gradients from previous iterations to predict gradient for subsequent iterations which in turn breaks the serial nature of standard stochastic optimization thereby enabling approximately parallel iterations. The gradient prediction is done using Kernel methods. The paper also presents experimental evidence towards the effectiveness of the proposed approach.

**Strengths:**

This appears to be an interesting contribution, albeit lacking clarity in terms of writing.

**Weaknesses:**

-- The question of gradient estimation converging to the true gradient seems fairly far-fetched, and there is every reason to believe modeling gradients would essentially carry a constant bias without very strong regularity and realizability conditions.

-- Furthermore, when applied to problems in RL, this becomes even more confounding because of the necessity to explore and using data collected with exploration to in turn construct gradient estimates. I am not sure if I follow how this can be achieved within this framework.

-- The paper's experimental comparison doesn't offer any sources of comparisons against the many different approaches to parallelization that have already been studied in the literature, e.g. mini-batching, model averaging etc.

**Questions:**

Can the authors clarify the speedup offered by utilizing their framework while using acceleration/momentum based methods? Standard mini-batch methods offer different trade-offs particularly wrt batch sizes used while still achieving linear parallelization speedups [1].

[1] Cotter et al: Better Mini-Batch Algorithms via Accelerated Gradient Methods.

**Limitations:**

N/A.

---

> ### Author Rebuttal · Authors · 2024-08-07
>
> We thank reviewer ieTR for taking the time to review our paper and appreciate the reviewer's feedback. We would like to provide the following response to address the concerns.
>
> ---
>
> > The question of gradient estimation converging to the true gradient seems fairly far-fetched, and there is every reason to believe modeling gradients would essentially carry a constant bias without very strong regularity and realizability conditions.
>
> Thank you for your insightful comment. We would like to clarify that Theorem 1 and Corollary 1 theoretically show that our kernelized gradient estimation **asymptotically** converges to the true gradient with respect to the number $T_0$ of gradient history, under our assumptions in Section 5. **This means the estimation error diminishes to nearly zero (not a constant bias) only when $T_0$ is sufficiently large.** If $T_0$ is large enough, thanks to the continuity of gradients, we typically is able to accurately approximate the gradient at any point in a local region. This also aligns with results in Bayesian optimization literature, where function estimation converges to the true function asymptotically without a constant bias, given enough function queries, as evidenced in [32].
>
> We acknowledge that in practice, obtaining a sufficiently large $T_0$ can be challenging, leading to biased gradient estimation. However, this biased estimation generally provides useful update directions, resulting in the acceleration rate $\Theta(\sqrt{N})$ that is achievable by our OptEx framework, as verified by our empirical results in Section 6.
>
> We hope this clarification addresses your concerns and demonstrates the robustness and potential of our approach.
>
> > Furthermore, when applied to problems in RL, this becomes even more confounding because of the necessity to explore and using data collected with exploration to in turn construct gradient estimates. I am not sure if I follow how this can be achieved within this framework.
>
> As mentioned in our Appendix B.2.2, our OptEx framework is built on the Deep Q-Network (DQN) algorithm for our RL experiments. In this context, we treat RL as a standard first-order optimization problem, where OptEx aims to accelerate the sequential gradient-based optimization of DQN using parallel computing. The exploration component, which is inherent in the DQN algorithm, does not require additional consideration within the OptEx framework. The exploration strategy used (i.e., the ε-greedy policy) inherently will primarily introduce noisier gradients (i.e., a larger $\sigma^2$), leading to more biased gradient estimates in our OptEx. Interestingly, our empirical results show that our OptEx is still able to achieve enhanced convergence even in the presence of this increased noise, demonstrating the effectiveness and robustness of our OptEx framework.
>
> > The paper's experimental comparison doesn't offer any sources of comparisons against the many different approaches to parallelization that have already been studied in the literature, e.g. mini-batching, model averaging etc.
>
> Our OptEx framework is in fact a complementary (or orthogonal) approach to the existing parallelization methods, aiming to speedup the first-order optimization in a more general way especially when the other parallelization methods are **not applicable** or **underperforming** rather than replacing them within the scenarios where they already work well (refer to our Sec. 2). For example, mini-batch and model averaging methods are typically used within the context of machine learning and may not be applicable for speeding up other optimization problems where batch sizes and model averaging are not possible, such as the synthetic function in Section 6.1. In contrast, our OptEx framework can provide a compelling speedup in these cases, as evidenced by the results in our Section 6.1. Even within the context of machine learning in our Sec. 6.3, when batch size is already sufficiently large, data parallelism (i.e., mini-batching and model averaging) no long can provide noticeable speedup to the optimization whereas our OptEx can make its novel contributions, as evidenced by the results in our Sec. 6.3 where a large batch size of 256/512 is applied. Overall, since OptEx and other parallelization methods are targeting for different scenarios, it is typically hard to make a fair experimental comparison among these methods.
>
> > Can the authors clarify the speedup offered by utilizing their framework while using acceleration/momentum based methods? Standard mini-batch methods offer different trade-offs particularly wrt batch sizes used while still achieving linear parallelization speedups [1].
>
> Thank you for your question. Our OptEx framework is designed to complement existing methods, including acceleration/momentum-based methods (refer to our Sec. 2). When combined with these methods, OptEx is still able to provide additional speedup by parallelizing iterations. For example, in our experiments on synthetic functions (Section 6.1), we used Adam as the baseline optimizer. Our OptEx framework further improved Adam's performance, achieving an acceleration of approximately $\sqrt{N}$ in practice. This demonstrates the superior effectiveness and wide applicability of OptEx when combined with other optimization techniques. We will make this clearer in our revised paper.
>
> ---
>
> We hope this clarification addresses your concerns and can increase your opinions of our work. We are happy to provide more clarifications.

---

> > ### Author Response · Authors · 2024-08-12
> >
> > Dear Reviewer ieTR,
> >
> > Thank you so much for taking the time to review our paper and for your insightful questions. While we have thoroughly addressed the concerns and questions raised by all the other reviewers, we sincerely hope that our clarifications and discussions above have effectively addressed your concerns as well.
> >
> > If you have any more questions or need more details, we are happy to answer them promptly within the discussion period.
> >
> > Best, Authors

---

### Official Review · Reviewer_RYzW · 2024-07-24

**Soundness:** 3
**Presentation:** 3
**Contribution:** 3
**Rating:** 7
**Confidence:** 3

**Summary:**

This paper introduces a new approach for parallelizing stochastic gradient descent for unconstrained, nonconvex, smooth stochastic optimization. The approach is based on building a Gaussian Process surrogate for the true gradient (based on a history of observed stochastic gradients), and uses this surrogate to suggest potential 'future' iterates, by iterating gradient descent using the surrogate. Iterates are then adjusted based on new stochastic gradient estimates, computed in parallel.

They present worst-case complexity theory that says that being able to compute $N$ stochastic gradients in parallel improves the iteration complexity by a factor of $\sqrt{N}$, with numerical results on a selection of synthetic problems, reinforcement learning problems, and neural network training problems.

**Strengths:**

1. I think the overall approach is an interesting contribution. I particularly like the discussion about the quality of the GP surrogate for the true function and think these results are interesting (Section 5.1). I think this could have useful links with optimization of genuinely complex functions, similar to those typically considered in Bayesian and zeroth order optimization.

2. The numerical results show some promise for the method, and do indeed show promise for a framework for using parallel stochastic gradient evaluations, where this is practical.

3. The paper is clearly written and easy to follow.

**Weaknesses:**

1. There are gaps in the literature survey of parallelized SGD methods. For example, there is no mention of the highly cited and relevant papers:
- Recht et al. Hogwild!: A Lock-Free Approach to Parallelizing Stochastic Gradient Descent. NeurIPS 2011
- Mnih et al. Asynchronous Methods for Deep Reinforcement Learning. ICML 2016
- Yu et al. Parallel Restarted SGD with Faster Convergence and Less Communication. AAAI, 2019

2. It is unclear to me whether the convergence theory (Theorem 2) actually supports the assertion of a $\sqrt{N}$ speedup from $N$ processes. The RHS gradient bound indeed is $O(1/\sqrt{N})$, but you are measuring the smallest gradient over $O(N)$ outer iterations. So, doesn't this say that you decrease your gradient by a factor of $O(1/\sqrt{N})$ after doing $N$ times extra work, which is exactly the convergence rate of SGD under your sub-Gaussian stochastic gradient assumption [33]? The work would also benefit from a direct comparison with just sample averaging of stochastic gradients (see question below).

3. It appears to me that Theorem 3 appears to contradict standard theory (e.g. Bottou, Curtis & Nocedal. Optimization Methods for Large-Scale Machine Learning. SIAM Review, 2018). You claim that the complexity bound for (nonconvex) SGD is tight when applied to the spherical function eq (47), which is strongly convex. However (stochastic) gradient descent with properly chosen stepsizes should converge linearly for this problem, not sublinearly as claimed. Can you explain this potential mismatch in results?

4. For very large problems, it is unrealistic that parallelism will be available, since storing the model and training data may already require parallelism (i.e. a single stochastic gradient evaluation is already computed in parallel). It would be helpful to include a comment about the specific regimes (dimension, dataset size, level of parallelism) where this method is intended to be used.

**Questions:**

How is the next iterate of OptEx defined? Line 10 of Algorithm 1 says it is the final iterate that is chosen, but Figure 1 suggests that the next iterate is somehow taken from a combination of the $\theta_t^1, \ldots, \theta_t^N$ values. Can you please clarify? The latter interpretation would make more sense, otherwise all but one of the parallel evaluations are discarded immediately.

Is the $O(1/\sqrt{N})$ speedup exactly the same as what you would get by just doing basic sample averaging? That is, at each iteration, just get $N$ different stochastic gradient estimates at the same point and average them (reducing the variance of the estimates). How does your bound compare with this simple use of parallelism?

You say that "$\nabla F$ is assumed to be sampled from the Gaussian Process, $\nabla F \sim GP(0, K(\cdot,\cdot))$". Can you please clarify? I read this to mean that this is the assumed prior of your GP surrogate for grad F, but as written it suggests that your actual objective function has - on average - zero gradient, meaning we would expect the 'average point' to be almost stationary?

You say the "Target baseline is equivalent to Algo. 1 with [GP mean] being replaced with [stochastic gradient], indicating the desired parallelized iteration we aim to approximate". Does the 'target' implementation also have the parallel step? If so, we would expect that to give an unfair advantage to 'target', since it does $2N$ stochastic gradient evaluations per outer iteration rather than $N$ for OptEx.

**Limitations:**

Limitations appear to have been addressed. The theoretical assumptions (e.g. smoothness) are clearly articulated and some practical limitations are addressed in Section 7.

No potential negative societal impact of this work.

---

> ### Author Rebuttal · Authors · 2024-08-07
>
> We thank Reviewer RYzW for recognizing the **interesting contribution, promise, and clarity** of our OptEx framework. We address your concerns below.
>
> ---
>
> ## Responses to Weaknesses
>
> 1. Thank you for pointing out this oversight. We appreciate you highlighting these relevant papers on data parallelism. We have referenced some of these works (e.g., [8]) in Section 2 to contrast iteration parallelization with data parallelism. We will include the additional related works you mentioned in our revised paper for a more comprehensive literature survey.
>
> 2. Thank you for your insightful comment. Measuring the smallest gradient over $N$ parallel processes is indeed **meaningful and necessary** because our OptEx framework aims to parallelize the $NT$ sequential iterations in SGD. To understand the speedup of OptEx, we follow the common practice in standard SGD to measure the smallest gradients over $NT$ updates (including $N$ parallel processes and $T$ sequential iterations) for a **clear and fair comparison** with standard SGD. Measuring the smallest gradient over $N$ does not decrease the gradient by a factor of $1/\sqrt{N}$. Instead, OptEx works by **decreasing the number of sequential iterations by a factor of $1/N$ with parallelism $N$** (refer to  the first term on the RHS of Equation 45).
>
> 3. Thank you for your thoughtful comment. We would like to clarify that **stochastic** gradient descent (SGD) indeed enjoys only a **sublinear** lower and upper bound (i.e., $\Theta(1/T)$ for strongly convex functions, as presented in [R1], where the convergence is measured by $E[\left\|x_t - x^*\right\|^2]$. In our paper, the gap between $\Theta(1/T)$ in [R1] and $\Theta(1/\sqrt{T})$ arises because we measure the convergence of OptEx using $\left\| \nabla F(x_t) \right\|^2$, in line with the measurement applied in our Theorem 2 for non-convex functions $F$. When using this measurement for convergence, the $\Theta(1/\sqrt{T})$ result in our paper **aligns perfectly** with the existing results for SGD as presented in [34].
>
>     [R1] Nguyen, Phuong_Ha, Lam Nguyen, and Marten van Dijk. "Tight dimension independent lower bound on the expected convergence rate for diminishing step sizes in SGD." Advances in Neural Information Processing Systems 32 (2019).
>
> 4. Thank you for the constructive suggestion. OptEx is designed to **complement**, **not replace**, existing parallel approaches (data, model, pipeline) for additional speedup when sufficient computing resources are available (Section 2). In resource-limited scenarios, simpler approaches should be prioritized due to OptEx's complexity. OptEx is most effective when our kernelized gradient estimation is accurate. Empirical results in Section 6.3 show OptEx performs well in **moderate-scale** optimization problems ($d \approx 10^6$, $N \leq 10$), achieving significant improvements. We will add these discussions to our revised paper to help practitioners assess OptEx's suitability.
>
> ## Responses to Questions
> 1. Thank you for pointing out this potential confusion. In our main paper, we choose the next iterate $\theta_t$ of OptEx to be $\theta_t^{(N)}$, where all the gradient evaluations ( $\{\theta_t^{(i)}\}_{i=1}^N$) are **not discarded**. These evaluations are used to construct our kernelized gradient estimation for the next iterate (Section 4.3), aiming to reducing estimation error (Theorem 1) and improve convergence (Appendix B.3). Figure 1 aims to suggest that all $N$ processes are **necessary** in each iterate of OptEx to ensure its effective performance. We will add these discussions to our revised paper to make this clearer.
>
> 2. The $O(1/\sqrt{N})$ speedup achieved by OptEx matches that of basic sample averaging for stochastic optimization with noisy gradients. However, the speedup from OptEx comes from **reduced sequential iterations** (first term on the RHS in Equation 45), while sample averaging derives from **reduced gradient variance** (second term on the RHS in Equation 45). When gradient noise is already small or in deterministic optimization (Section 6.1), data parallelism may not provide noticeable speedup, but OptEx can still contribute significantly. Overall, OptEx works in a complementary direction to existing parallelization methods, including sample averaging, to speed up first-order optimization, especially when other methods are not applicable or underperforming (Section 2).
>
> 3. Thank you for your question. Similar to the common practice in the Bayesian optimization literature [21], $\nabla F \sim GP(0, k(\cdot, \cdot))$ means that $\nabla F$ can be regarded as **being sampled from** a GP prior $GP(0, k(\cdot, \cdot))$ (refer to line 213), which is only a prior for $\nabla F$ and can not infer $\nabla F=0$ as $\nabla F$ can be **any** function within this prior. Given an increasing number of gradient histories, we will gain a better understanding of $\nabla F$, i.e., the GP posterior mean can gradually better approximate $\nabla F$, as evidenced by our Thm.1. We will add these discussions to our revised paper to clarify this point.
>
> 4. Thank you for your question. The 'target' implementation does not have the parallel step due to the inherent iterative dependency in first-order optimization. We treat 'target' as an **ideal, yet impractical**, iteration parallelization (refer to Appendix B.1). This baseline highlights the gap between the ideal and our OptEx, showing that OptEx achieves an acceleration of $\sqrt{N}$ instead of $N$. We will add these discussions to our revised paper for clarity.
>
> ---
>
> With our elaboration and clarification, we hope our response has addressed your concerns and increased your opinions of our work. We are happy to provide more clarifications if needed.

---

> > ### Comment · Reviewer_RYzW · 2024-08-08
> >
> > Thank you for these clarifications, they are very helpful.
> >
> > Pending seeing a new version of the manuscript, I think the responses given for all points above would be very helpful. I have one remaining concern (question Q2) and one point for clarification (weakness Q2) that I still think warrant more consideration.
> >
> > **Questions #2**
> >
> > I understand that you would like to potentially use OptEx in the exact gradient case ($\sigma=0$, e.g. Section 6.1), and your results make sense in this context. However, in the SGD context that seems to be your main focus, if I had the ability to compute $N$ stochastic gradients in parallel (each with a given noise level $\sigma$ as you assume), I could either use OptEx or compute $N$ stochastic gradients at the current iterate and average these, giving an effective variance of $\sigma^2/N$.
> >
> > In Theorem 1, your complexity bound (when $\sigma>0$) is $O(\sigma/\sqrt{NT})$ for $T$ iterations, which suggests that sample averaging applied to standard SGD (where $\sigma \mapsto \sigma/\sqrt{N}$) would achieve the same complexity as OptEx with essentially no overhead from managing the GP, etc.
> >
> > It would be useful if you could either discuss why/when OptEx should be preferred to simple sample averaging, as I'm not sure if your theoretical results show this. See, for example, the discussion "Trade-Offs of (Mini-)Batching" in Section 4 of Bottou, Curtis & Nocedal [your ref 26].
> >
> > **Weaknesses #2**
> >
> > I think my confusion arose because your statement of Theorems 2 and 3 are not quite correct. Specifically, the LHS term in eqs (8) and (9) should be $\min_{\tau\leq T, s\leq N} \|\nabla F(\theta_{\tau,s})\|^2$ rather than $\min_{\tau\leq NT} \|\nabla F(\theta_{\tau})\|^2$ as given (i.e. reflecting that you do $T$ outer iterations with $N$ parallel evaluations in each, rather than $NT$ outer iterations with $N$ parallel evaluations in each, as currently stated). This seems to align with what is shown in the proofs.

---

> ### Author Response · Authors · 2024-08-08
>
> Dear Reviewer RYzW,
>
> Thank you so much for your positive feedback and thoughtful suggestions! We sincerely appreciate your recognition and support. We are glad to hear that our clarifications are very helpful. We will add our clarifications above in the writing to make our paper clearer. We would like to address your remaining concerns below.
>
> ---
>
> ### Questions #2
>
> We totally agree with you that sample averaging applied to standard SGD can achieve similar complexity as our OptEx, without the overhead of managing the GP. As we have justified above, these two speedup approaches stem from different principles:
>
> - OptEx Speedup: This comes from reduced sequential iterations (first term on the RHS in Equation 45).
> - Sample Averaging Speedup: This is derived from reduced gradient variance (second term on the RHS in Equation 45).
>
> So, when gradient noise $\sigma$ is already decreased to a small value through sample averaging with parallelism $N'$ (not $N$), increasing $N'$ further (i.e., $N'+N$) provides diminishing returns due to the sublinear rate of $O(1/\sqrt{N'+N})$. For example, the benefit of reducing from $1/\sqrt{10}$ to $1/\sqrt{15}$ is less significant compared to reducing from $1/\sqrt{1}$ to $1/\sqrt{6}$ with an additional parallelism $N=5$.
>
> However, in this scenario of a small gradient noise $\sigma$, our OptEx can still achieve noticeable speedup (e.g., from $1/\sqrt{1}$ to $1/\sqrt{6}$ with the same additional parallelism $N=5$) by leveraging additional parallel computing to reduce the sequential iterations. This reduction in sequential iterations is however unattainable by mini-batch SGD itself.
>
> In light of this, Equation 45, leading to our Theorem 2, in fact, will provide valuable insights into when OptEx is preferable to simple sample averaging. Overall, OptEx complements rather than replaces existing parallelization methods, including sample averaging, to accelerate first-order optimization. This is particularly effective when other methods are **inapplicable** or **underperforming**, i.e., further noticeable improvements cannot be achieved merely by increasing parallel computing resources in those methods, as we have justified above.
>
> ### Weaknesses #2
> We apologize for the confusion caused by our notation $\tau$. Our intention was to use $\tau$ to denote all gradients evaluated during the optimization process more easily, including those in parallel processes, as mentioned in line 260. In Theorems 2 and 3, the expression $\min_{\tau \leq NT} \left|\nabla F(\theta_{\tau})\right|$ is exactly meant to represent $\min_{t \leq T, s \leq N} \left|\nabla F(\theta_{t,s})\right|$. Thank you for bringing this to our attention. We will correct this notation in our revised paper.
>
> ---
>
> We want to thank Reviewer RYzW for your positive feedback and recognition again! We sincerely hope our response has addressed your remaining concerns and can increase your opinions of our work. If you have any other questions, we would like to address them.
>
> Sincerely, Authors

---

> > ### Comment · Reviewer_RYzW · 2024-08-09
> >
> > Thanks for this information. Regarding question #2, I think this is a very important aspect and warrants a discussion in any new version of the paper. However, I don't think the benefit is as good as suggested: if you want to do $N_1$ parallel samples for each stochastic gradient (to reduce the variance to $\sigma^2/N_1$), then if you run OptEx with $N_2$ parallel runs, then you will need $N_1\times N_2$ stochastic gradients in parallel at each outer loop, not $N_1+N_2$, so your speedup is still $O(1/sqrt{N})$ where $N=N_1\times N_2$ is the number of cores needed.
> >
> > This has been a helpful discussion, and pending the clarifications in a revised version as agreed by the authors above, I am happy to upgrade my scores of the paper in the initial review above.

---

> ### Author Response · Authors · 2024-08-09
>
> Dear Reviewer RYzW,
>
> Thank you so much for your prompt and thoughtful response. We are pleased to hear that our clarifications have improved your opinion of our work, and we will incorporate all the clarifications and discussions mentioned above into our revised manuscript, as per your suggestion.
>
> We also appreciate your valuable insight regarding the parallel computing required by combining sample averaging and OptEx, in relation to your question #2. We agree with you that in this scenario, we will need $N=N_1 \times N_2$ parallel computations, leading to a speedup of $O(1/\sqrt{N})$. The benefits of our OptEx over sample averaging may be more straightforward by looking into our Equation 45, which mirrors the convergence results of sample averaging. Particularly, when the gradient noise $\sigma$ is already decreased to a small value, i.e., the second term on the RHS in Equation 45 is small, the first term on the RHS in Equation 45 that is related to sequential iterations will then dominate the convergence. Consequently, simply applying sample averaging with increased parallel computational resources yields only marginal improvements in convergence. In contrast, OptEx can achieve a noticeable improvement by reducing the first term on the RHS in Equation 45 by $O(1/N)$. We will also include this discussion in our revised paper. Should you have any other questions, we would like to address them.
>
> Sincerely, Authors

---

### Decision · Program_Chairs · 2024-09-25

**Decision:**

Accept (poster)

**Comment:**

The paper presents OptEx, a first-order optimization framework that enhances the efficiency of first-order optimization algorithms (FOO) through approximately parallelized iterations. OptEx mitigates FOO's reliance on numerous sequential iterations by using kernelized gradient estimation to predict required gradients based on historical data, allowing for approximate parallelization.

The paper establishes theoretical guarantees for the estimation error of the kernelized gradient method and the iteration complexity of SGD-based OptEx, showing that the estimation error approaches zero as gradient history accumulates. Numerical examples are provided to showcase the performance of the method in various settings.

Overall, most reviewers find the paper’s contributions to be novel, interesting, and valuable. Some questions, concerns, and suggestions have arisen during the review process, which the authors have adequately addressed. They are strongly encouraged to incorporate these into the final version of the paper.